# Controlling the Risk of Corrupted Contexts for Language Models via Early-Exiting

**Andrea Wynn** [1]  **Metod Jazbec** [2]  **Charith Peris**[* 3]  **Rinat Khaziev**[* 4]
**Anqi Liu** [1]  **Daniel Khashabi** [1]  **Eric Nalisnick** [1]

## Abstract

Large language models (LLMs) can be influenced by harmful or irrelevant context, which can significantly harm model performance on downstream tasks. This motivates principled designs in which LLM systems include built-in mechanisms to guard against such "garbage in, garbage out" scenarios. We propose a novel approach to limit the degree to which harmful context can degrade model performance. First, we define a baseline "safe" behavior for the model – the model's performance given no context at all (zero-shot). Next, we apply distribution-free risk control (DFRC) to control the extent to which the user-provided context can decay performance below this safe zero-shot baseline. We achieve this by leveraging dynamic early exit prediction, ignoring later attention heads that attend the most to the unsafe inputs. Finally, we propose modifications to DFRC that allow it to both control risk for harmful inputs *and* leverage performance and efficiency gains on helpful inputs. We present both theoretical and empirical results across 9 tasks spanning in-context learning and open-ended question answering, showing that our approach can effectively control risk for harmful context and simultaneously achieve substantial computational efficiency gains with helpful context.

## 1. Introduction

Large language models (LLMs) have shown an impressive ability to be adapted to a wide variety of tasks through methods such as prompt tuning and in-context learning, which adapt the model with minimal data by manipulating its context rather than through expensive fine-tuning. Yet this adaptability introduces safety concerns: incorrect, misleading, or otherwise harmful context can degrade performance or elicit unsafe outputs. Imagine an LLM deployed for a specific use case and adapted through user-provided context; this context may be misleading for a number of reasons – such as unintended user error, malicious intent, or task mis-specification. For instance, consider an LLM assisting clinicians with summarizing patient notes. A busy nurse might accidentally provide an incorrectly labeled demonstration, marking a non-urgent case as urgent due to time pressure. Alternatively, a biased clinician may omit or skew important details due to their perception about a particular patient. These kinds of inconsistent, low-quality contexts can shift the model's behavior toward unreliable triage recommendations. Such errors could escape the notice of a human system designer, motivating us to develop built-in safeguards so that the LLM defaults to disregarding these compromised inputs.

In this paper, we apply distribution-free risk control (DFRC) to mitigate the influence of corrupted context by comparing the loss of the adapted model to the zero-shot model. LLMs' zero-shot performance on a wide variety of tasks has become quite strong in recent years and continues to improve (Kojima et al., 2023). This makes zero-shot LLMs comparatively well-understood and predictable, whereas in-context learning (ICL) or context-influenced models on arbitrary user-supplied inputs may reflect uncontrolled or adversarial distribution shifts. Using the zero-shot model as a baseline also anchors risk control in a setting that has undergone extensive pre-deployment safety testing (Zhang et al., 2024; Yuan et al., 2025), unlike the highly variable user-provided contexts that arise during deployment. Recent work has shown that depth controls how much an LLM can learn from its context; LLMs "overthink" on harmful inputs, meaning their performance peaks at some intermediate layer and drops in deeper (later) layers (Halawi et al., 2024). We replicate these results on our tasks; an example is shown in Fig. 1(b), where the model does well given helpful context, reasonably well given no context (zero-shot), and poorly given harmful context. Inspired by this, we implement early-

---

[*]This work does not relate to the author's position at Amazon. [1]Johns Hopkins University [2]University of Amsterdam [3]Amazon AGI [4]Amazon Alexa. Correspondence to: Andrea Wynn <awynn13@jhu.edu>.

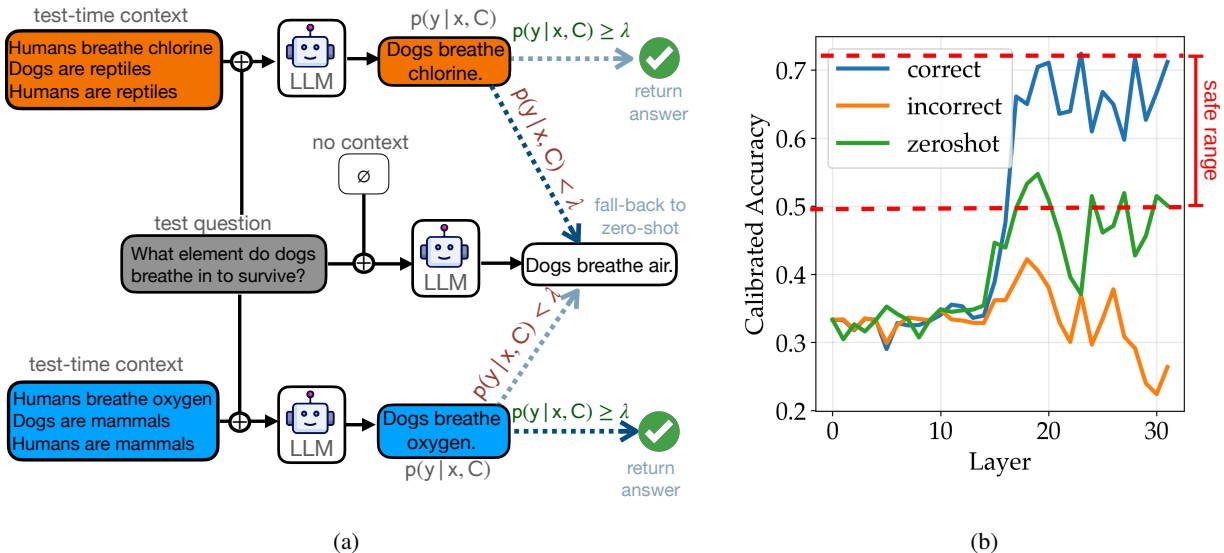

(a)                                           (b)

*Figure 1.* (a) A LLM is given some input context C of unknown quality (helpful or harmful). For a test question x, the model needs to infer whether to rely on the answer it obtains using the given context without knowing ahead of time if it was helpful or not. If not, it falls back to the answer it would give without seeing any additional context (zero-shot). (b) When given harmful context (such as incorrect in-context demonstrations), it is better to either early-exit or simply ignore the given context than to use the model's final prediction – staying in the "safe" performance range between zero-shot and correct (or helpful) context.

exiting as a mechanism for applying risk control.

In summary, we propose three important contributions to ensure safety under arbitrary context: a novel formulation of early-exit models for safety (§3.1), a novel context-aware loss designed to measure overthinking (§3.3), and an important adaptation of the Learn-then-Test (LTT) risk control framework balancing safety with efficiency (§3.4). We show via extensive experiments that our approach prevents over-thinking while still allowing the LLM to benefit from helpful context, providing robust safety guarantees even with mixed-quality inputs (§4.3) and enabling major computational efficiency gains compared to prior approaches (§4.4). We conduct experiments across 9 diverse tasks and 5 distinct models, spanning a wide variety of in-context learning settings and an open-ended, multi-token question answering task. Our framework guarantees safety on model outputs relative to zero-shot while simultaneously achieving a greater than 50% speedup in comparison to previous approaches across all tasks. Overall, to the best of our knowledge, this is the first work to establish a principled framework controlling the risk of potentially harmful user-provided context, while simultaneously leveraging dynamic early exit mechanisms to achieve performance and computational efficiency gains with helpful context.

All code needed to reproduce our results can be found at https://github.com/andreawynn/controlling-corrupted-context-risk.

## 2. Preliminaries

**Data** Let $x \in \mathcal{X}$ denote an input text (e.g., a question), and let $y \in \mathcal{Y}$ be its associated label (e.g., the associated answer). Given our focus on answer prediction, the label space is denoted by the set $\mathcal{Y}$, which may represent a finite set of valid classes (for classification) or an unbounded set of answers to a question. In the classification setting, $\mathcal{Y}$ is finite and can be defined as $\mathcal{Y} = \{1, \ldots, K\}$, with $K$ representing the number of possible answers. Moreover, we denote the *context* provided by the user to the model as $c$. In the case of in-context learning, $c$ can also be represented as a *context set* of $N_c$ demonstrations as $c_{\text{ICL}} = \{(x_i, y_i)\}_{i=1}^{N_c}$. The data-generating distribution over $\mathcal{X} \times \mathcal{Y}$ is denoted $\mathcal{P}$.

**Model** We denote by $p(\cdot|x, c)$ an LLM that takes a given input $x$ together with the context $c$ and outputs a probability distribution over possible labels in $\mathcal{Y}$. To construct a set of demonstrations $c$ for in-context learning, we sample both the input $x$ and the context set $c_{\text{ICL}}$ from the same dataset, but explicitly prevent any overlap between $x$ and the elements of $c_{\text{ICL}}$ for a given prompt. By excluding $x$ from $c_{\text{ICL}}$, we can prevent the model "copying" answers from the context rather than demonstrating meaningful in-context learning behavior, while still drawing both from the same distribution to avoid any bias in context/input dataset selection.

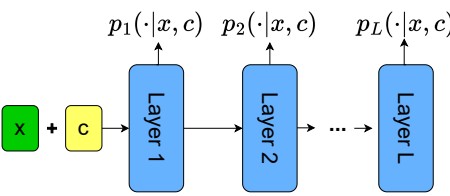

$p_1(\cdot|x,c)$  $p_2(\cdot|x,c)$  $p_L(\cdot|x,c)$

*Figure 2.* With an early-exit LLM, we execute every layer until our confidence exceeds the $\lambda$ threshold, after which we directly make a prediction from the intermediate layer.

$$\hat{y}_\lambda := \begin{cases} \arg\max\limits_{k\in\mathcal{Y}} p_1\left(k|x,c\right) & \text{if } \mathfrak{C}_1 \geq \lambda, \\ \arg\max\limits_{k\in\mathcal{Y}} p_2\left(k|x,c\right) & \text{else if } \mathfrak{C}_2 \geq \lambda, \\ \vdots & \vdots \\ \arg\max\limits_{k\in\mathcal{Y}} p_L\left(k|x,c\right) & \text{otherwise.} \end{cases} \quad (1)$$

### 2.1. Early-Exiting for Classification Tasks

Traditionally, LLMs pass through all layers of the model before making a prediction. In contrast, early-exit LLMs (Elbayad et al., 2020; Schuster et al., 2022) offer the option to yield a prediction after each layer. This is achieved by passing the current hidden representation through an "unembedding" matrix that maps the hidden state to the vocabulary, pruned to the finite set of possible labels for classification tasks. Specifically, for each layer $l \in \{1, \ldots, L\}$ where $L$ represents the total number of layers in the model, a confidence score $\mathfrak{C}_l \in [0, 1]$ and an exit threshold $\lambda \in [0, 1]$ are defined. An early prediction is returned as soon as the confidence at the current layer exceeds the threshold, evaluating the conditions sequentially from layer 1 to $L$, as outlined in Eq. 1 and illustrated in Fig. 2.

Here, $p_l$ denotes the LLM's predictive distribution at the $l$-th layer. While various choices of confidence scores are possible, we use a simple one derived from the maximum class probability: $\mathfrak{C}_l := \max\limits_{k\in\mathcal{Y}} p_l(k \mid x, c)$. This choice of confidence measure is common in prior work (Schuster et al., 2022), and we provide additional justification of this choice (compared to alternative confidence scores) through ablation studies in §G.2.

### 2.2. Controlling Predictive Risk via Distribution-Free Risk Control

Risk control frameworks (Angelopoulos et al., 2022; Bates et al., 2021) enable principled selection of thresholds $\lambda \in \Lambda$ across various machine learning problems, ranging from conformal prediction (Angelopoulos et al., 2024) to adaptive inference (Schuster et al., 2022; Jazbec et al., 2024). Concretely, first a problem-specific *loss* function $\ell : \mathcal{Y} \times \mathcal{Y} \to \mathbb{R}$ is defined. The *risk* associated with a candidate threshold $\lambda$

is then defined as the expected loss

$$R(\lambda) := \mathbb{E}_{(x,y)\sim\mathcal{P}}\left[\ell(p_\lambda(x), y)\right]$$

where $p_\lambda$ denotes a threshold-dependent predictor (e.g., an early-exit LLM, see Eq. 1). The goal is to leverage a calibration dataset $\mathcal{D}_{\text{cal}} \sim \mathcal{P}^{N_{\text{cal}}}$ to find a threshold $\hat{\lambda}$ such that the risk is guaranteed to be small – i.e., $R(\hat{\lambda}) \leq \epsilon$ for some $\epsilon > 0$ – on new test points, which are assumed to be independently and identically distributed, or *iid*, with the samples from $\mathcal{D}_{\text{cal}}$.

## 3. Context-Aware Risk Control via Early-Exit

In this section, we detail our approach to mitigating overthinking under potentially harmful context, i.e., preventing the LLM from picking up on harmful context by combining early-exiting with risk control. We begin by introducing a novel formulation of early-exit models for safety by using the safe zero-shot model as a baseline (§3.1). We then show that applying our early-exit approach can effectively prevent overthinking while still allowing the LLM to benefit from helpful context, provided an appropriate exit threshold $\lambda$ is chosen (§3.2). Next, we propose a novel context-aware loss designed to measure overthinking, on which we can apply risk control to ensure safety on new test points (§3.3). Finally, we introduce a simple adaptation of the Learn-then-Test (LTT) framework to accommodate losses that may take on negative values, as is often the case when both helpful and harmful context is present in the data (§3.4).

### 3.1. Safe Context-Aware Predictor

We begin with our base model $p$ given input $x$ and context $c$ without early-exit, $p(\cdot|x,c)$. This model can "overthink" (overfit to potentially harmful context $c$), significantly degrading the output over the zero-shot model, $p(\cdot|x)$. We propose to define a new safe context-aware early exit model $\bar{p}_\lambda(\cdot|x,c)$ by augmenting this base model in the following two ways: (i) enable the model to make predictions from intermediate exits given a confidence threshold and (ii) if at no exit the confidence exceeds the threshold, ignore the context $c$ and use the zero-shot prediction. With these augmentations, the safe context-aware model is defined as:

$$\bar{y}_\lambda := \begin{cases} \hat{y}_\lambda & \text{if } \exists i \ s.t. \ \mathfrak{C}_i \geq \lambda, \\ \arg\max\limits_{k\in\mathcal{Y}} p_L\left(k|x\right) & \text{otherwise} \end{cases} \quad (2)$$

where $i \in \{1, ..., L\}$. Such a predictor enables us to leverage early-exit to gain efficiency and performance on helpful context, while also intervening early to avoid harmful context before the model fully processes it. When early-exit alone is insufficient to guarantee safety, *the zero-shot model serves as a reliable baseline* (see the last condition in Eq.2).

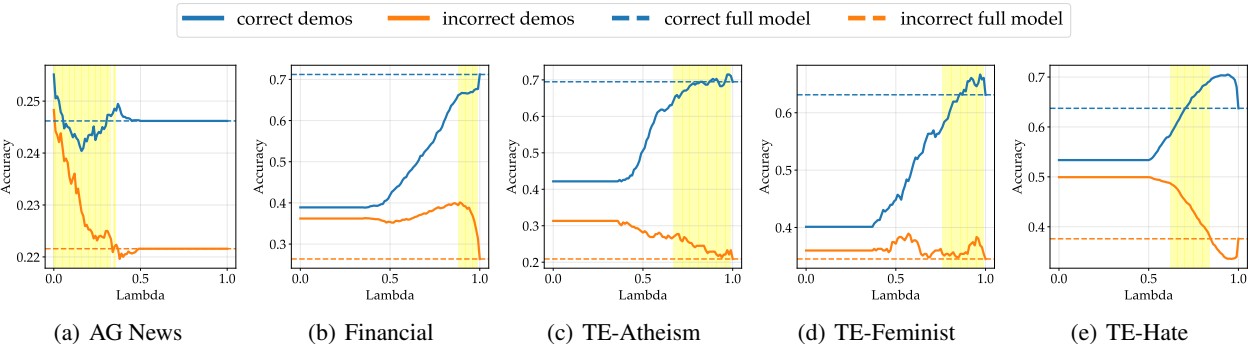

*Figure 3.* Some choices of $\lambda$ thresholds can both attain performance gains from helpful context *and* control overthinking from harmful context. The highlighted regions show where $\lambda$ values exist such that **we lose no more than 5% of the accuracy gains from correct (helpful) in-context demonstrations while still doing better than the full model given incorrect (harmful) in-context demonstrations**.

## 3.2. Early Exit Reduces Overthinking

As observed by (Halawi et al., 2024), overthinking primarily arises in the deeper (i.e., later) layers of models, where it can override the strong inductive biases or correct predictions established in earlier layers. This phenomenon can result in degraded or unsafe outputs, as illustrated in Fig. 1. While prior work proposes pruning certain attention heads in the later layers as a preventive measure (Halawi et al., 2024), we instead advocate for leveraging dynamic early exiting (Teerapittayanon et al., 2016) which has previously been employed to mitigate overthinking in settings outside the scope of LLMs and in-context learning (Kaya et al., 2019; Jazbec et al., 2023). Notably, early exiting offers a natural solution to the overthinking problem: by terminating inference at an intermediate layer, it prevents the model from fully processing potentially misleading context. Thus, stopping early increases accuracy when the context is harmful. We show that overthinking behavior occurs on all our tasks in Fig. 10. Although not our primary goal, a notable side benefit of early-exiting is efficiency gains, as not all layers are processed before outputting a prediction.

We demonstrate the effectiveness of early exiting in mitigating overthinking on harmful context in Fig.3. Across most of our datasets and for a broad range of exit thresholds $\lambda$, an early-exit LLM ($p_\lambda$) outperforms the full model ($p_L$) on inputs with harmful context (§D). Importantly, for certain thresholds – the yellow highlighted area in Fig.3 – early termination also does not significantly degrade performance on samples with helpful context. These results underscore the potential of dynamic inference to prevent LLMs from picking up on harmful context.

## 3.3. Context-Aware Prediction Risk

After describing how early exiting can be used to prevent overthinking, we now turn to the problem of selecting an appropriate early-exit strategy using risk control, as introduced

in §2.2. As a first step, we propose a novel context-aware prediction loss:

$$\ell_c(\lambda; x, y, c) := \ell(\bar{y}_\lambda(x, c), y) - \ell(\hat{y}(x), y), \quad (3)$$

where $\bar{y}_\lambda(x, c) := \arg\max_k \bar{p}_\lambda(k|x, c)$ and $\hat{y}(x) := \arg\max_k p_L(k|x)$ denote the predictions of the safe context-aware model with context $c$ (see Eq. 2) and the full zero-shot model, respectively. Crucially, both predictions are produced by the same underlying LLM. The $\ell_c$ loss compares the performance of the early-exit model with context to that of the model without context. This formulation makes it well-suited for measuring overthinking: if the context is harmful, the loss will be positive; if it is helpful, the loss will be negative. In contrast, prior early-exiting work (Schuster et al., 2022; Jazbec et al., 2024) has focused exclusively on loss definitions that compare early-exit outputs to final-layer outputs given the same input. Such losses are less effective for identifying and addressing overthinking, as they do not account for the drop in performance in later layers due to harmful context. Our model thus allows for robust measurement of risk considering *both helpful and harmful context*, enabling risk control with mixed context.

Having defined the overthinking loss, we now turn to identifying an appropriate threshold $\hat{\lambda}$ using a suitable risk control framework and a calibration dataset $\mathcal{D}_{cal} = \{(x_i, y_i, c_i)\}_{i=1}^{N_{cal}}$. Our goal is to find a threshold for which the overthinking risk is small, i.e.,

$$R_c(\hat{\lambda}) = \mathbb{E}_{(x,y,c)\sim\mathcal{P}}[\ell_c(\hat{\lambda}; x, y, c)] \leq \epsilon,$$

where $\epsilon > 0$ is a user-specified tolerance level representing the acceptable degree of overthinking. Naturally, smaller values of $\epsilon$ impose stricter control, prioritizing thresholds that suppress overthinking more aggressively. However, this may come at the cost of reduced performance on helpful context—a tradeoff we explore in §F.3. Since we observe that risks computed using the context-aware loss $\ell_c$ are not monotonic with respect to $\lambda$ (Fig. 9), the Learn-then-Test

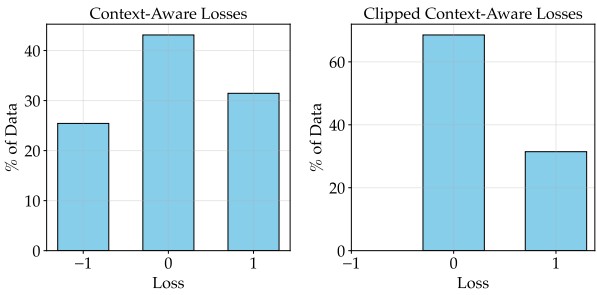

*Figure 4.* We show the distribution of our context-aware loss on the TweetEval-Hate dataset with a 50% mix of helpful and harmful context. There are a significant number of negative loss values, which the loss-clipping approach sets to 0. Our risk transformation approach enables us to preserve the original underlying loss distribution.

(LTT) framework ([Angelopoulos et al., 2022](#)) is the only viable option; hence, we use LTT as our risk-control approach for selecting our exit threshold $\hat{\lambda}$.

### 3.4. Domain-Preserving Risk Transformation

While LTT ([Angelopoulos et al., 2022](#)) supports non-monotonic losses/risks, it requires the loss to be bounded, $\ell \in [0, 1]$, due to its reliance on the Hoeffding-Bentkus bound ([Bentkus, 2004](#)). ([Schuster et al., 2022](#)) circumvented this by clipping all negative loss values to zero. However, for our context-aware prediction risk, negative losses are important: they correspond to helpful context from which we want to leverage performance gains over zero-shot. These negative losses are also quite common on our tasks, and clipping them means we lose substantial information about the true underlying loss distribution (Fig. 4). By clipping these losses to zero, the risk control procedure cannot distinguish between performing *at* or *better than* the baseline (in our case, zero-shot). This leads to substantially more conservative early-exiting, which can make LTT impractical when we want to favor helpful context for performance improvement and greater efficiency gains.

We propose a novel risk transformation approach to overcome this limitation on LTT. In particular, given a risk level $\epsilon$ and a bounded loss $\ell(\lambda; x, y, c) \in [a, b]$ for any $a, b \in \mathbb{R}$, we can execute the following procedure:

1. Compute $\epsilon' = \frac{\epsilon - a}{b - a}$ and $\ell'(\lambda; x, y, c) = \frac{\ell_c(\lambda; x, y, c) - a}{b - a}$.

2. Define $\epsilon'$ as the new risk level and $\ell'(\lambda; x, y, c)$ as the new loss. $\ell'(\lambda; x, y, c) \in [0, 1]$ and $\epsilon' \in [0, 1]$, so the Hoeffding-Bentkus bound is satisfied.

3. Apply LTT to select $\hat{\lambda}$ with the risk level $\epsilon'$ and the risk $R(\ell'(\hat{\lambda}; x, y, c))$. $\hat{\lambda}$ also controls risk $R(\ell(\hat{\lambda}; x, y, c))$ at level $\epsilon$, as we prove in §B.

The key insight for this approach to work is that controlling

the risk $R(\lambda) := \mathbb{E}_{x,y,c}[\ell]$ at level $\epsilon$ is equivalent to controlling the risk $R'(\lambda) := \mathbb{E}_{x,y,c}[\ell']$ at level $\epsilon'$. Intuitively, this is because we are applying the same invertible transformation to both the loss and the risk level, and we can simply reverse the transformation after applying LTT to return to the domain of the original loss and risk level. The full proof can be found in §B.

**Application to Context-Aware Risk Control.** On all our tasks, our context-aware prediction loss is bounded by $\ell_c(\lambda) \in [-1, 1]$; hence we can plug in $a = -1, b = 1$ when performing loss scaling in our setting. We contrast our approach with the previous approach of clipping negative losses to zero $\ell_{\text{clip}}(\lambda) := \max\{0, \ell_c(\lambda)\}$ and empirically verify that our approach both satisfies the same risk control guarantees and achieves much greater efficiency gains across all tasks (see Fig. 5 and 7 for in-context learning and Fig. 6 and 8 for open-ended QA).

## 4. Experiments

### 4.1. In-Context Learning

**Tasks** We use a total of 8 in-context learning tasks for our work, spanning three diverse domains (sentiment analysis, hate speech detection, and semantic classification). Our in-context learning tasks are the Stanford Sentiment Treebank ([Socher et al., 2013](#)), FinancialPhrasebank ([Malo et al., 2013](#)), TweetEval-Hate, -Atheism, and -Feminist ([Barbieri et al., 2020](#)), AG News ([Del corso et al., 2005](#)), Text REtrieval Conference (TREC) ([Li & Roth, 2002](#)), and Unnatural ([Halawi et al., 2024](#)). A detailed description of these datasets and the domains they cover can be found in §A.1.

**Models** We compare four models – two regular LLaMA models (Llama-3-8B ([Grattafiori et al., 2024](#)) and Llama-2-7B ([Touvron et al., 2023](#))) and two LayerSkip LLaMA models (layerskip-llama3-8B and layerskip-llama2-7B ([El-houshi et al., 2024](#))). The LayerSkip models are additionally pre-trained to encourage the production of higher-quality intermediate representations, which provide a helpful point of comparison with the models that are not explicitly pre-trained for early-exit. We provide further details on our experiments in §A.2.

### 4.2. Open-Ended Question Answering

**Motivation.** While our ICL experiments demonstrate risk control on classification tasks, many real-world LLM deployments involve open-ended, multi-token generation where users provide context documents. For instance, returning to our clinical scenario: a physician might upload patient notes (context) and ask the LLM to generate a treatment recommendation. If those notes contain errors or inconsistencies, we may want the LLM to recognize this and fall back to more conservative zero-shot behavior. We evaluate

this setting using SQuAD v2.0, which includes both helpful context (containing the correct answer to the corresponding question) and adversarially-selected harmful context (chosen to mislead the model into giving an incorrect answer) on a question-answering (QA) task.

**Task & Model.** We experiment with SQuAD v2.0 (Rajpurkar et al., 2018), which includes questions with helpful context that include the correct answer to the corresponding question and questions with harmful context that are adversarially selected by human annotators to mislead the model, containing reasonable but incorrect answers. We use the T5-Large model (Raffel et al., 2020), mirroring the approach taken in CALM (Schuster et al., 2022). We provide further details on our experiment design in §A.4.

### 4.3. Empirical Verification of Risk Control Guarantees

We empirically verify that our approach always controls the risk across all models and datasets, even when given a mix of helpful and harmful context. Fig.5 and 6 show that our approach satisfies the DFRC guarantees on risk: the context-aware risk is controlled across all models, tasks, and risk levels $\epsilon$. We provide additional results in the appendix (§E) comparing these results with the loss-clipping approach, showing that our risk transformation approach is consistently less conservative and better matches the user-defined risk level $\epsilon$ than loss-clipping approach. This highlights both the validity of our theoretical results as well as its applicability to real-world tasks. We additionally demonstrate that our risk-control guarantees hold regardless of the distribution of helpful vs harmful context (§F.2).

**Safety of the Zero-Shot Baseline on Language Modeling Experiments.** Our zero-shot fallback mechanism demonstrates selective intervention on unsafe inputs. At our risk-controlled thresholds $\hat{\lambda}$, the system defaults to zero-shot predictions 3.3x more frequently on unanswerable (misleading) questions than answerable ones (3.0% vs. 0.9%), indicating that our approach successfully discriminates on context quality. Further, when fallback occurs with unsafe (unanswerable) context, zero-shot outperforms early-exit predictions on 71% of the questions, confirming that intervention happens precisely when needed. This demonstrates a substantial safety benefit of our approach compared to CALM, which defaults to using full-model predictions that have processed the corrupted context.

**Class-Conditional Effects.** We examine the effects of our chosen $\hat{\lambda}$ on different sub-populations of our data, corresponding to whether the model is given helpful or harmful context. We find that where there exists a $\hat{\lambda}$ that both controls overthinking risk and preserves the accuracy gains from helpful context, our method is able to find it. In other cases, we find that there can be a direct tradeoff between taking advantage of performance gains given helpful context and

controlling overthinking risk given harmful context, depending on the choice of $\hat{\lambda}$; results are shown in the appendix in Fig.21. We further describe this tradeoff in §F.3.

### 4.4. Comparison of Efficiency Gains with Loss Clipping

Our adaptation of the loss-bounding approach outperforms prior LTT approaches (Jazbec et al., 2024; Schuster et al., 2022) by increasing efficiency gains while preserving the same risk control assurances. At all $\epsilon$ levels, our efficiency gains – the number of layers of computation we save by applying our approach – are strictly greater than when using clipped risk (Fig. 7 & 8), and are often significantly greater. For example, controlling the prediction gap risk with our approach at $\epsilon = 0.05$ on our in-context learning tasks results in an average of $53\%$ fewer layers evaluated over all datasets and models compared to loss-clipping. We achieve even more significant efficiency gains on the open-ended question answering experiments; on SQuAD v2.0, we evaluate up to $81.8\%$ fewer layers. Though not a primary goal of our work, it is a notable benefit of our approach.

## 5. Related Work

**Learning from Context.** In-context learning (ICL), also known as few-shot learning, enables LLMs to perform new tasks by conditioning on a limited number of input-output examples within the prompt without requiring gradient-based fine-tuning (Dong et al., 2024; Brown et al., 2020; Srivastava et al., 2023). Methods like ICL allow LLMs to adapt to novel tasks by generalizing from context provided in the prompt. However, this flexibility also makes LLMs vulnerable, as not all context provided to a model improves performance. Novice users who provide incorrect examples of a task may make an LLM perform worse on a task than it would have without the user's input (Halawi et al., 2024), or adversarial users can design prompts to make LLMs bypass their safeguards (Xu & Wang, 2024). In our work, we consider how the *quality* of in-context demonstrations can impact the safety of the model's outputs.

**Evolution of representations through LLM layers.** Recent studies have begun mapping how LLM hidden representations evolve with depth, revealing structured processing phases. (Cheng et al., 2025; Cheng & Antonello, 2024; Cheng et al., 2023) identify a pronounced intermediate-layer "abstraction phase" in which hidden states predict brain responses to language stimuli, showing that LLMs compress inputs into low-dimensional manifolds early in processing. Layer-wise analysis has also been applied to tasks such as ciphers (Fang et al., 2025), long-context failures (Lu et al., 2024), and multilingual representations (Bafna et al., 2025; Muller et al., 2021; Wendler et al., 2024; Schut et al., 2025) in LLMs. We leverage these intermediate representations to determine when to safely make a prediction on a task.

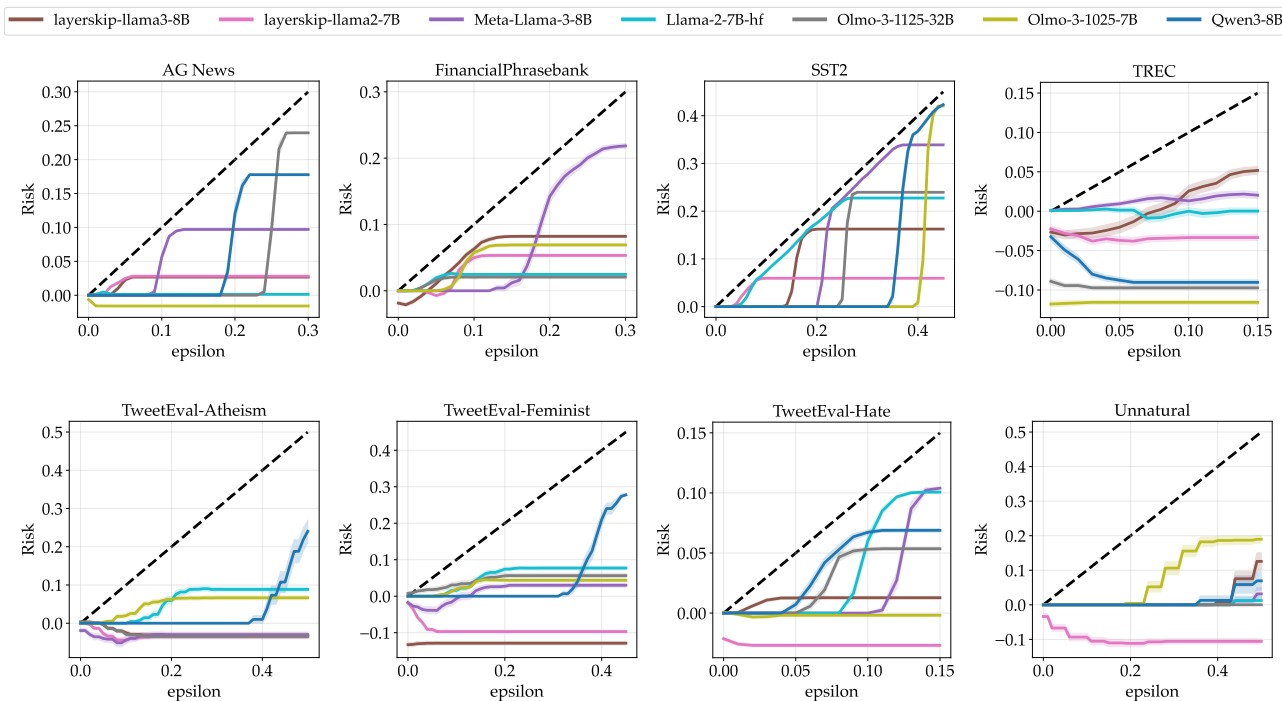

*Figure 5.* Empirical risk vs the user-specified risk level $\epsilon$ using our safe context-aware model and $\ell_c$ loss over a set of mixed correct and incorrect demonstrations on our ICL tasks. Aligning with the theoretical guarantees, the risk is controlled across all models and tasks. Shaded regions correspond to one standard error over 100 experiments and are included on all plots.

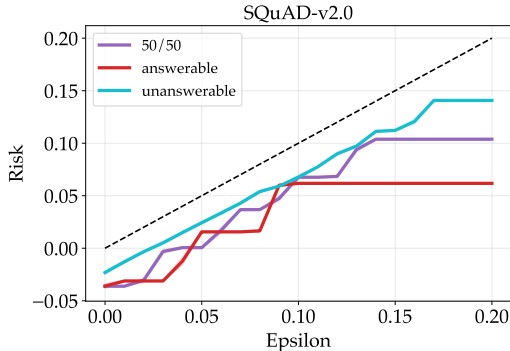

*Figure 6.* Results of applying our risk control approach on SQuAD-v2.0 using the CALM model on different mixtures of context quality. Shaded regions correspond to one standard error over 100 experiments and are included on all plots. We demonstrate that our risk transformation approach effectively balances performance gains from helpful context with safety on misleading context while preserving robust risk-control guarantees.

**Distribution-free risk control (DFRC).** Risk control is a statistical framework for controlling various measures of risk in machine learning systems. Given a trained model, a finite set of calibration data, and a loss function reflecting the chosen measure of safety, DFRC bounds the expected loss, i.e., *risk*, as a function of some low-dimensional parameter $\lambda$ (Angelopoulos et al., 2024; Bates et al., 2021). Among

existing frameworks, Learn Then Test (LTT) (Angelopoulos et al., 2022) is widely used, as it is the only method that provides guarantees without requiring monotonicity of the loss or risk. However, LTT assumes the loss is bounded within $[0, 1]$, which can be restrictive in certain scenarios (see §3.4 for a discussion on the implications of bounded loss in the ICL setting and our approach to addressing this limitation). Notable examples of using risk control in the context of LLMs include controlling performance degradation due to accelerated inference (Schuster et al., 2022; Jazbec et al., 2024) and mitigating prompt-induced variability (Zollo et al., 2024). In contrast, our work is the first to leverage risk control for managing the impact of harmful demonstrations on the downstream performance of LLMs.

**Early Exiting.** Early exiting in deep neural networks enhances computational efficiency by terminating inference at intermediate layers for simpler inputs, thereby reducing resource consumption with minimal performance degradation (Teerapittayanon et al., 2016). While early exiting has been widely adopted to accelerate inference (Huang et al., 2018; Zhou et al., 2020; Elbayad et al., 2020; Han et al., 2021; Schuster et al., 2022; Jazbec et al., 2024), our work introduces a novel application of early exit architectures: mitigating the influence of harmful context provided to a language model. Although the use of early exiting to prevent ICL overthinking has also been discussed in (Halawi

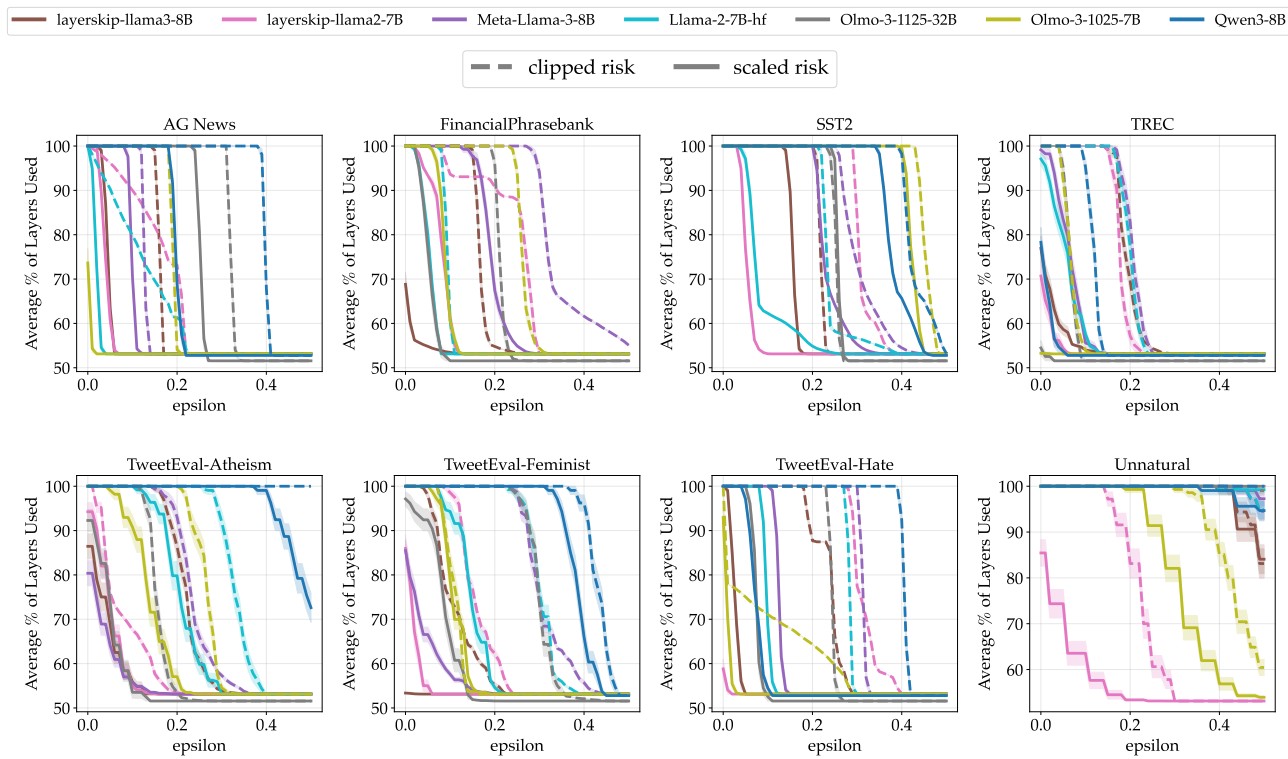

*Figure 7.* We demonstrate that our risk transformation approach enables much greater efficiency gains than the loss-clipping approach by leveraging performance gains from correct demonstrations in ICL.

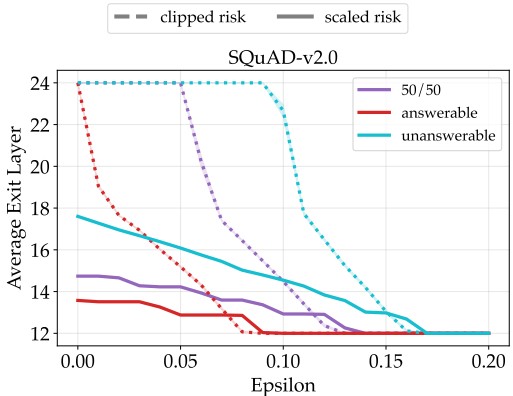

*Figure 8.* We demonstrate that our risk transformation approach enables much greater efficiency gains than the loss-clipping approach by leveraging performance gains from helpful context on the SQuAD dataset with different mixtures of context quality.

et al., 2024), their approach relies on static layer pruning. In contrast, our method employs dynamic, per-sample early exiting based on confidence thresholding, which enables us to make robust guarantees for safety via risk control.

# 6. Conclusion and Discussion

Our work introduces a novel risk-controlled early-exit framework for safely and robustly leveraging user-provided context of mixed quality. We present three important contributions to enable safety in this setting: a novel early-exit model formulation using a zero-shot baseline; a novel context-aware prediction loss designed to measure overthinking; and an adaptation of the LTT risk control framework to work for our setting. This integrated approach improves the safety, reliability, and computational efficiency of learning from mixed-quality context.

Our work makes design choices specific to our tasks and models that may require adaptation in other settings. First, we prevent early-exit in the first half of layers, since early layers are frequently overconfident in the wrong answers (§G.3); while empirically validated across all our tasks and models, the optimal cutoff may differ in other scenarios. Second, our per-token threshold $\lambda$ is well-suited to the short-answer QA setting of SQuAD (§G.5), but may be less effective for longer-generation tasks. For instance, a model generating a long harmful response might be high-confidence on only one or two intermediate tokens; in such cases, averaging $\lambda$ over all tokens may be preferable.

However, while these specific design choices in our frame-

work may require adaptation as models and tasks evolve, the core challenge of detecting and intervening on harmful outputs mid-generation will remain a fundamental concern; evidence that intermediate representations reliably encode harmful intent even under adversarial pressure (Zou et al., 2024) suggests that this class of approaches will continue to be viable even on future, more advanced models.

**Future work.** We leave systematic exploration of specific design choices, such as those listed above, to future work. More broadly, promising directions include: developing methods robust to validation/test distribution mismatch; extending our framework to handle rare but high-stakes safety violations; integrating our framework with complementary approaches such as early-stopping during reasoning (Wang et al., 2026), which performs analogous early-exit within the reasoning token sequence rather than at the architectural level; and investigating class-conditional risk control for more robust safety assurances under mixed-quality prompts (§F.3). Finally, future work could explore settings where helpful and harmful context co-occur within the same prompt; a practically important scenario given that real-world users may produce inconsistent prompt quality even for the same underlying task.

## Impact Statement

This work contributes to safer deployment of LLMs by introducing a principled method to detect and mitigate "overthinking" by accounting for potentially harmful context input to LLMs. Our approach additionally improves computational efficiency, making LLM inference more environmentally and economically sustainable. However, though our approach improves safety in the average case, it does not provide strong guarantees for specific worst-case prompts, potentially leaving some harmful scenarios unmitigated. Additionally, the increased reliance on automated risk control mechanisms may give users a false sense of security, especially if deployed without proper monitoring or human oversight.

ACKNOWLEDGMENTS

This research was conducted outside of Amazon and funded by a faculty award from the Johns Hopkins + Amazon Initiative for Interactive AI[1]. We acknowledge the use of computational resources from the Johns Hopkins DSAI cluster.

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

# A. Experimental Setup

## A.1. Detailed Description of In-Context Learning Tasks

**Sentiment analysis.** Sentiment analysis refers to the computational study of opinions, emotions, and attitudes expressed in text (Kumar et al., 2023), which requires inferring polarity or stance from often subtle or domain-specific cues. We use three sentiment analysis datasets: Stanford Sentiment Treebank (SST-2) (Socher et al., 2013) involves binary sentiment classification of movie reviews, Financial Phrasebank (Malo et al., 2013) extends this task to the financial domain, and TweetEval-Feminist (Barbieri et al., 2020) centers on sentiment detection toward feminism in social media posts.

**Hate Speech Detection.** Hate speech detection involves identifying language that expresses hatred, discrimination, or hostility toward individuals or groups, often within socially sensitive contexts. We examine two datasets from the TweetEval benchmark (Barbieri et al., 2020) that address this problem in distinct but related ways. The TweetEval-Hate dataset consists of tweets directly annotated for the presence or absence of hate speech, whereas the TweetEval-Atheism dataset is used in studying hate speech due to its focus on religion-related discourse, where antagonistic or prejudiced language is common.

**Semantic Classification.** Semantic classification tasks involve assigning text to high-level conceptual categories based on meaning, structure, or subject matter, focusing on identifying the core informational content of input text. We examine three semantic classification tasks: AG News (Del corso et al., 2005) involves classifying news headlines into four broad areas – World, Sports, Business, and Science/Technology. The Text REtrieval Conference (TREC) dataset (Li & Roth, 2002) involves classifying open-domain questions into six semantic types (e.g., entity, location, numeric). The Unnatural dataset, a toy dataset constructed by (Halawi et al., 2024), assigns short text descriptions to one of three semantic categories: sports, animals, or plants.

## A.2. In-Context Learning Experiment Design

We summarize our design as follows for the in-context learning tasks:

1. *Selecting Calibration Data:* From each dataset, we randomly draw 50% for our calibration dataset (on which we compute $\hat{\lambda}$) and the remaining 50% is our test data on which we present results.

2. *Label Transformation:* Existing datasets are often memorized during model pre-training (Li et al., 2024). Thus, we transform the tasks into a format that is equivalent to, but distinct from, their original form to mitigate these linguistic biases, mirroring the approach taken in prior work (Fang et al., 2025; Pan et al., 2023). We show that dataset memorization happens, and that this label transformation approach mitigates this effect, in §G.4.

3. *In-Context Demonstrations:* During the risk control calibration step, we compute a single $\hat{\lambda}$ for risk control on a 50-50 mix of *both incorrect and correct demonstrations*. Incorrect demonstrations are obtained by permuting the labels, as in (Halawi et al., 2024).

4. *Contextual Calibration:* Since our focus is on classification tasks, we examine how frequently the model assigns a higher probability to the correct label than to any alternative. However, model outputs can be highly sensitive to minor changes in the prompt (Gao et al., 2021), an observation which we verified through additional experiments, detailed in §G.4. To address this instability, we apply contextual calibration (Zhao et al., 2021) to balance the label probabilities.

5. *Evaluation Metrics:* We evaluate our models primarily on the in-context learning risk (as defined in §3.3), demonstrating that with our approach, risk always remains below the user-defined $\epsilon$ threshold.

## A.3. In-Context Learning Prompt Format

The prompt format is presented below for the AG News dataset. The same format is used across all in-context learning datasets, with the only difference being the list of possible labels. {text} indicates the input on which the model is asked to make a prediction. {demo i} and {label i} indicate the text-label pairs that constitute the in-context examples (where the label either corresponds to the true label or the substituted "incorrect" label).

**List of "dummy" labels.** We define a fixed substitution between the true labels of each dataset and the "dummy" labels we use in our prompts. In particular, for each true label, we substitute each label with one of the following words: river, stone, cloud, chair, table, grass. We find that there is not a significant effect of using any particular substitution, so we simply randomly select label-substitute pairs.

**Zero-Shot Prompt.**    Your job is to classify the topic of a news article given a description of the article. The possible topics are: world, sports, business, science/technology. Output only the topic of the news article and nothing else. Do not provide chain of thought reasoning before your answer. Description: {text} Topic:

**In-Context Demonstrations.**    Your job is to classify the topic of a news article given a description of the article. The possible topics are: world, sports, business, science/technology. Output only the topic of the news article and nothing else. Do not provide chain of thought reasoning before your answer. Below are a few examples of description-topic pairs. Description: {demo 1} Topic: {label 1} Description: {demo 2} Topic: {label 2} ... Description: {text} Topic:

**Dummy Labels - Zero-Shot Prompt.**    Your job is to classify the topic of a news article given a description of the article. Output river if the topic is world, stone if the topic is sports, cloud if the topic is business, and chair if the topic is science/technology. Output only the topic of the news article and nothing else. Do not provide chain of thought reasoning before your answer. Description: {text} Topic:

**Dummy Labels - In-Context Demonstrations.**    Your job is to classify the topic of a news article given a description of the article. Output river if the topic is world, stone if the topic is sports, cloud if the topic is business, and chair if the topic is science/technology. Output only the topic of the news article and nothing else. Do not provide chain of thought reasoning before your answer. Below are a few examples of description-topic pairs. Description: {demo 1} Topic: {label 1} Description: {demo 2} Topic: {label 2} ... Description: {text} Topic:

### A.4. Open-Ended Question Answering Experiment Design

We summarize our experiment design as follows for the SQuAD-v2.0 task:

Mirroring our in-context learning setting with incorrect and correct demonstrations, we separate the SQuAD-v2.0 questions into harmful (unanswerable) vs helpful (answerable) context. For our zero-shot model, we simply feed the questions without any context into T5-Large without any early-exiting. Since the SQuAD-v2.0 dataset does not have a list of valid answers for the unanswerable questions, we generate up to 5 answers for each of these questions with GPT-5 to measure model correctness.

During the risk control calibration step, we compute a single $\hat{\lambda}$ for risk control on a 50-50 mix of *both helpful and harmful context*. We use the same in-context learning risk as defined in 3.3 to evaluate our model compared to the user-defined $\epsilon$ threshold. On any example where the confidence does not exceed the $\lambda$ threshold at any layer across all tokens generated, we default to the zero-shot model for safety; we justify this strategy in §G.5.

## B. Proof of Risk Transformation Approach

Here, we prove that our risk transformation approach chooses an appropriate $\hat{\lambda}$ that controls our risk.

**Proof.**    Let $\ell(\lambda, x, c, y) \in [a, b]$, and choose a risk level $\epsilon \in (a, b)$ and some probability $\delta$. Compute $\epsilon' = \frac{\epsilon - a}{b - a}$ and $\ell' = \frac{\ell(\lambda, x, c, y) - a}{b - a}$. Apply the Learn Then Test procedure to $\epsilon'$, $\ell'$ to select $\hat{\lambda}$. Formally, as shown in (Angelopoulos et al., 2022), this guarantees that $\mathbb{P}(R(\ell') \leq \epsilon') \geq 1 - \delta$ for any chosen $\delta \in (0, 1)$. So, with probability at least $1 - \delta$, we have that $R(\ell') \leq \epsilon'$.

Towards contradiction, assume that $\hat{\lambda}$ does not control the original risk $R(\ell(\lambda, x, c, y))$ at level $\epsilon$. Thus, $R(\ell(\hat{\lambda}, x, c, y)) = \mathbb{E}_{x,y,c}[\ell(\hat{\lambda}, x, c, y)] > \epsilon$. By definition, $\ell = (b - a)\ell' + a$, so $\mathbb{E}x, y, c[\ell] = (b - a)\mathbb{E}x, y, c[\ell'] + a = (b - a)R(\ell') + a$. Similarly, by definition of $\epsilon'$, we have that $\epsilon = \epsilon'(b - a) + a$. So we can show the following:

$$R(\ell(\hat{\lambda}, x, c, y)) > \epsilon$$
$$R(\ell')(b - a) + a > \epsilon'(b - a) + a$$
$$R(\ell')(b - a) > \epsilon'(b - a)$$
$$R(\ell') > \epsilon'$$

However, we know that $\hat{\lambda}$ controls $R(\ell')$ at level $\epsilon'$. This is a contradiction. This proves that $\hat{\lambda}$ must also control

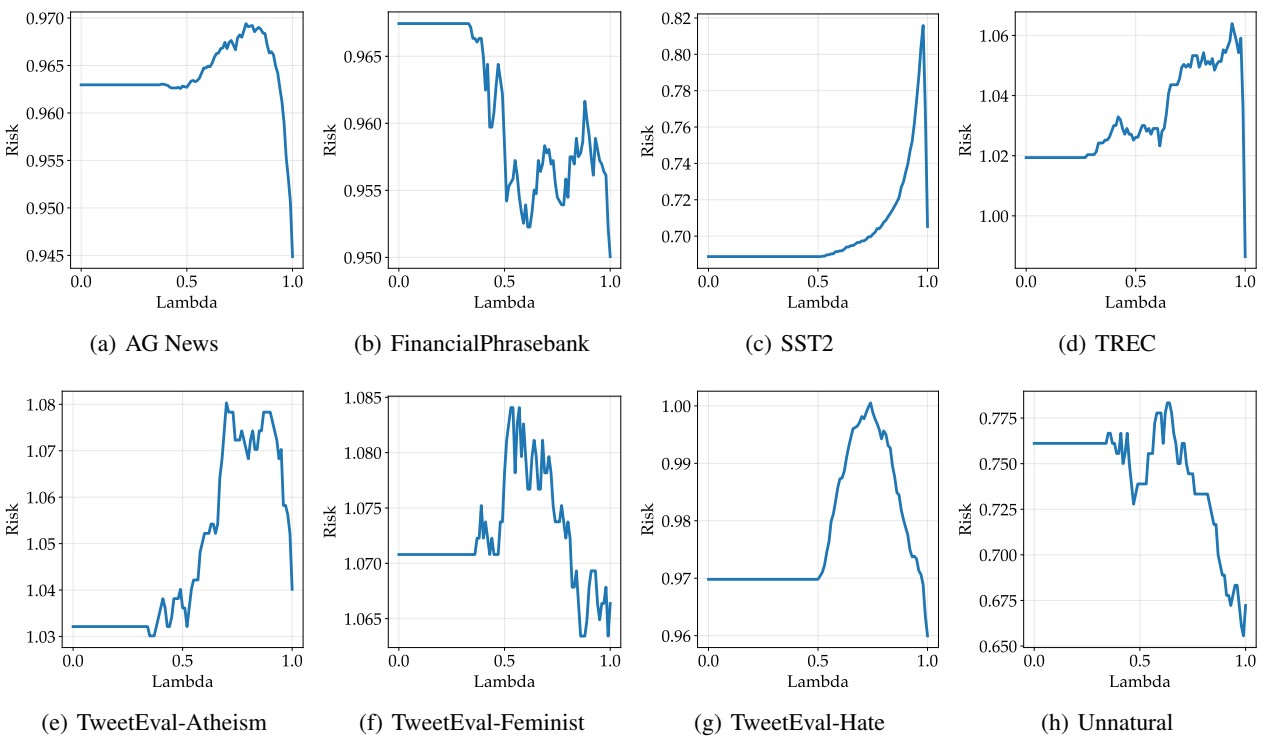

*Figure 9.* We show that across many of our models and datasets, the risk is non-monotonic in $\lambda$.

$R(\ell(\hat{\lambda}, x, c, y))$ at level $\epsilon$ with probability at least $1 - \delta$.

## C. Non-Monotonicity of Risk

We show that our risk is non-monotonic in $\lambda$ across many of our models and datasets in Fig.9. This indicates that we cannot use many of the existing methods in the conformal risk control literature, because they require an assumption of monotonicity (Jazbec et al., 2024), and motivates our choice of Learn Then Test in our work as it does not require this assumption.

## D. Overthinking Across Datasets

We show that overthinking occurs across all of our datasets in Fig.10. This provides additional motivation for using early-exiting as a natural approach to control risk on all tasks.

## E. Comparison of Risk Control Approaches

Across all tasks and models, we find that our risk transformation approach is less conservative and better matches the user-defined risk level $\epsilon$ than the loss-clipping approach. A direct comparison is illustrated in Fig. 11 with a mix of 50% correct and 50% incorrect demonstrations.

## F. Robustness of Risk Control Approach

### F.1. Demonstrating Robustness of Risk Control with Variation in Threshold $\lambda$

We provide a plot, Figure 12, with confidence intervals added to our results in Fig. 3 by bootstrapping samples from the datasets. We then only highlight $\lambda$ values for which no point within our confidence interval loses more than 5% of the accuracy gains from correct demonstrations while still doing better than the full model given incorrect demonstrations. We

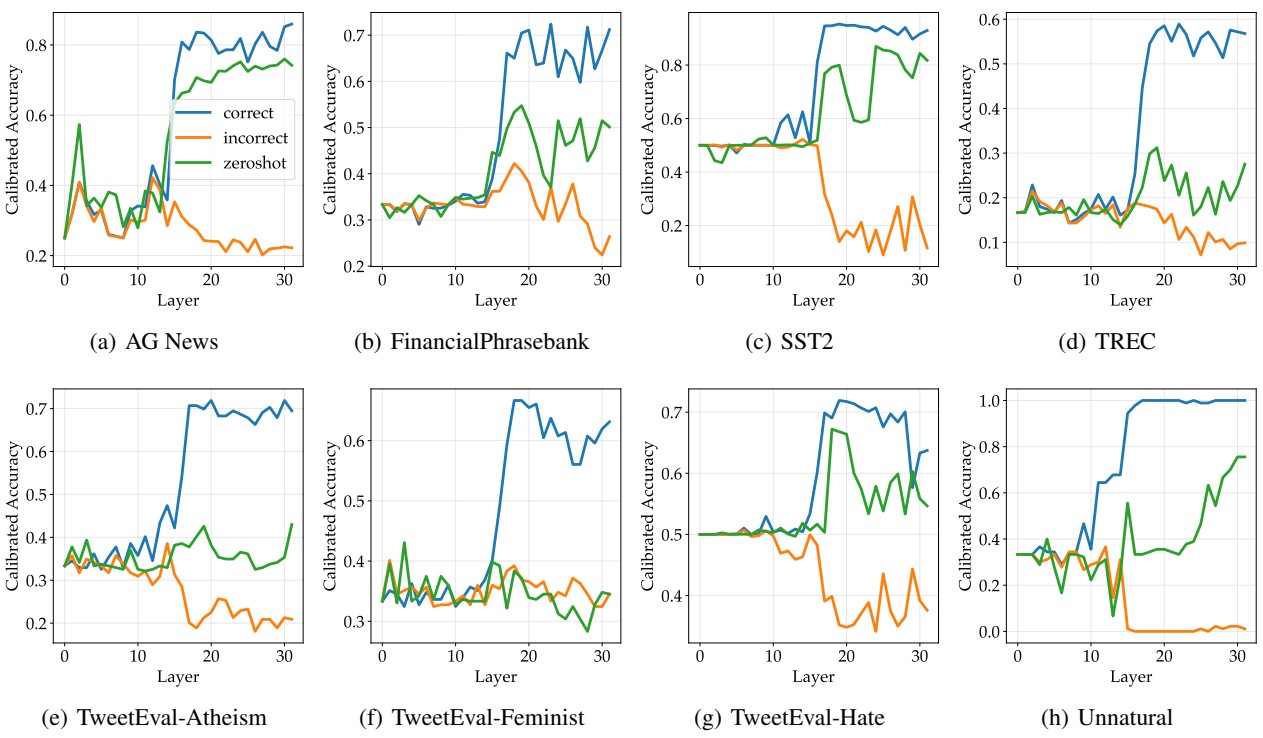

(a) AG News      (b) FinancialPhrasebank      (c) SST2      (d) TREC

(e) TweetEval-Atheism      (f) TweetEval-Feminist      (g) TweetEval-Hate      (h) Unnatural

*Figure 10.* Overthinking occurs across widely varying datasets - demonstrating that, if given incorrect in-context demonstrations, we should either early-exit or default to zero-shot behavior to ensure safety. However, if we are given correct demonstrations, we would like to both take advantage of the performance benefit (relative to zero-shot) and early-exit when we do not need all layers of the model to arrive at the correct answer. All plots are generated using the LayerSkip LLaMA-3 8B model.

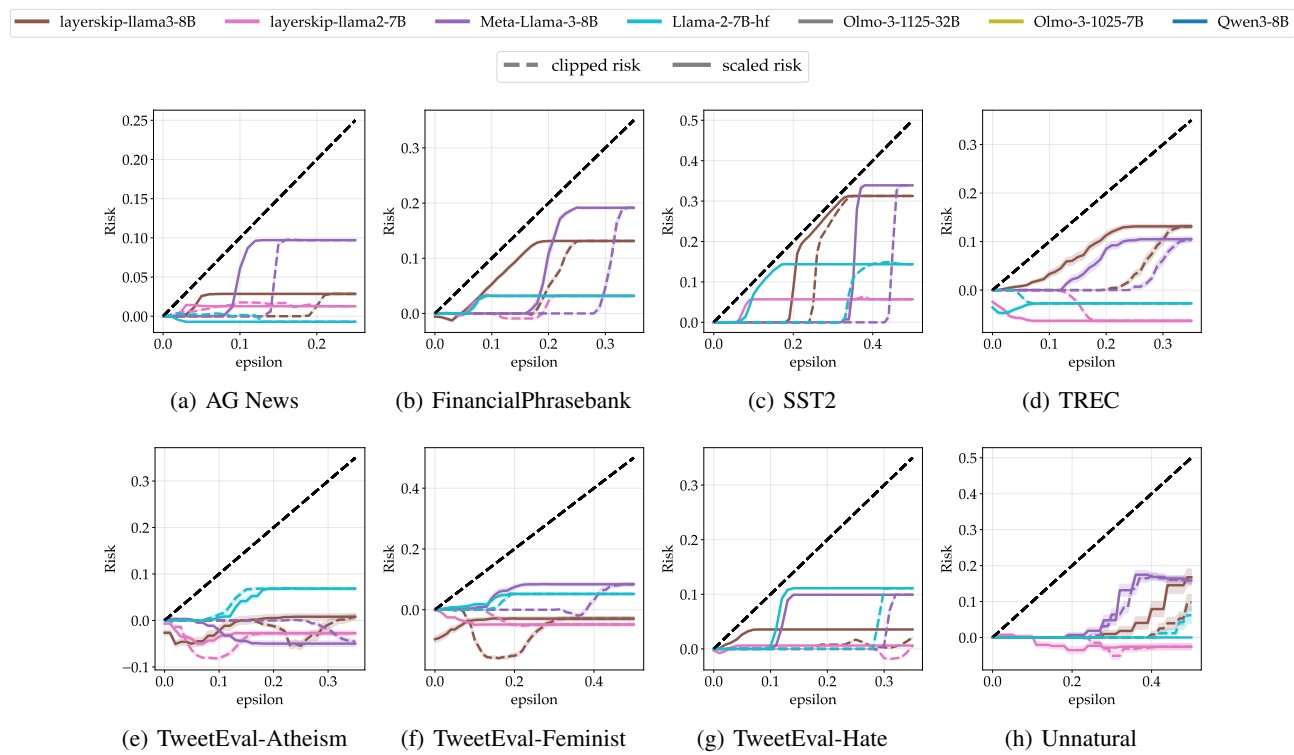

*Figure 11.* We show that across all tasks and models, our risk transformation approach is less conservative and better matches the user-defined risk level $\epsilon$ than the loss-clipping approach. These experiments used 50% correct and 50% incorrect demonstrations.

also include error bars for the accuracy of correct and incorrect demonstrations. Notably, even with these confidence bounds, there remain regions where the yellow band persists—demonstrating that even when accounting for the variance in samples, there still exist values of $\lambda$ that satisfy our desiderata. This provides clearer evidence for the robustness of our results.

## F.2. Risk Control at Any Proportion of Correct vs Incorrect Demos

Here, we show results demonstrating that regardless of the proportion of helpful vs harmful context in the calibration data, we are still able to control the combined risk over a test set drawn i.i.d. from the same distribution. Results shown in Figures 13, 14 and 15 for cases when there are more correct than incorrect demonstrations in ICL (a scenario that is likely in real-world applications), but we also show in Fig.16 that even when we have many more incorrect than correct demonstrations, our risk-control guarantees still hold. We provide similar results on our CALM open-ended multi-token QA task in Fig. 17 and 18, once again showing that we control the risk more efficiently under varying distributions of context quality.

## F.3. Class-Conditional Risk Control

We provide all results from our experiments investigating the class-conditional risk levels over all datasets and models using a 50-50 split of correct and incorrect demonstrations. Results are shown in Figures 21 and 20.

Though the marginal guarantees from risk control do not extend in theory to class-conditional risk control (how well our risk control works within each subgroup of our data, i.e. correct vs incorrect demonstrations) when using a mixed calibration dataset with both incorrect and correct in-context demonstrations, our approach still shows promise for finding an appropriate $\hat{\lambda}$ when one exists that *can* perform well for both correct and incorrect demonstrations. In practice, for applications where controlling risk given unsafe prompts is more important than performance gains from helpful prompts or achieving greater efficiency gains, we can instead use the loss-clipping approach, which is much more conservative and therefore may better control risk on unsafe prompts alone, as shown in Fig.19 and 20. However, even with loss clipping, there are no guarantees for risk control on incorrect demonstrations alone for the same reason; as shown in Fig.19 and 20, loss clipping can still

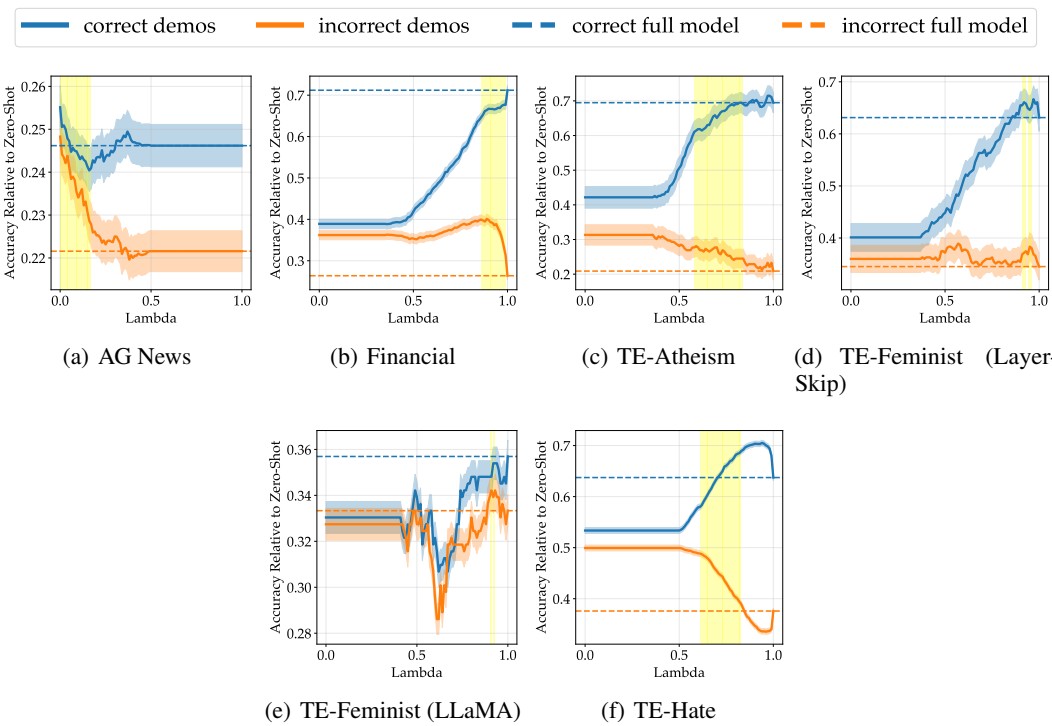

*Figure 12.* Some choices of $\lambda$ thresholds can both attain performance gains from correct demonstrations *and* control overthinking from incorrect demonstrations. We show the robustness of our approach to different selections of $\lambda$ by adding error bars that further restrict the choices of $\lambda$. Collecting more i.i.d. samples from the dataset will simply yield narrower error bars and thus more choices of $\lambda$.

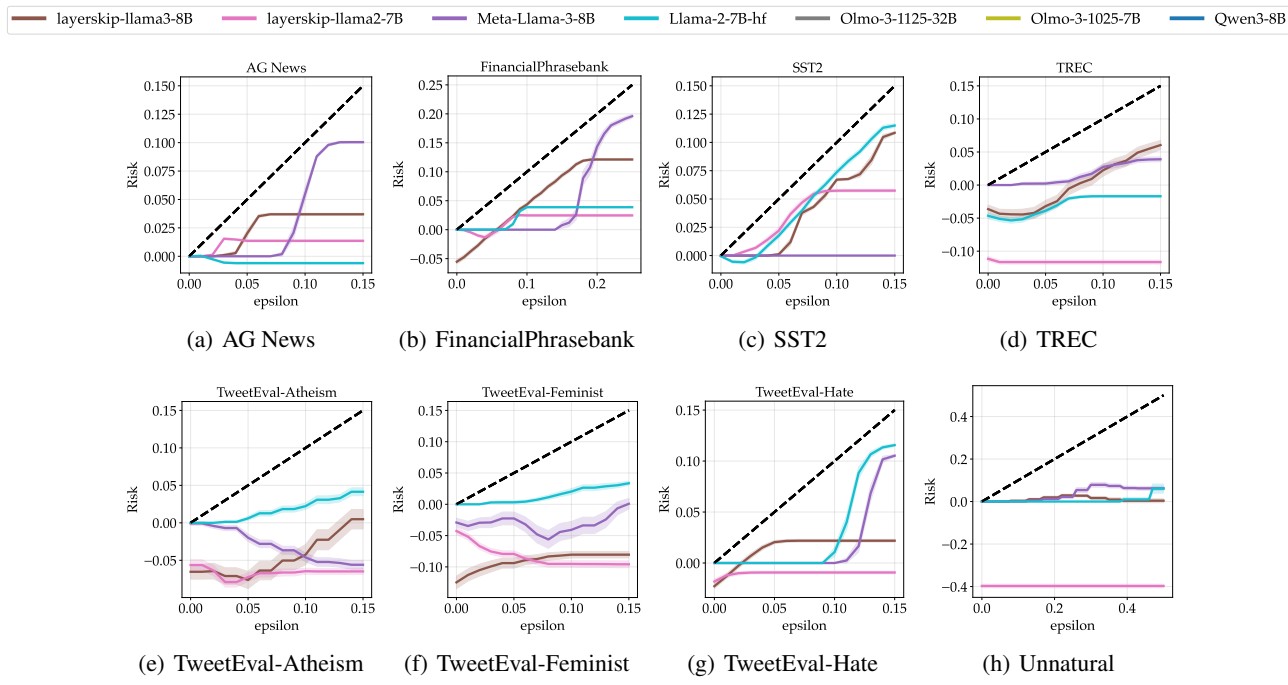

*Figure 13.* Empirical risk vs the user-specified risk level $\epsilon$ using our risk transformation approach over a set of 75% correct and 25% incorrect demonstrations.

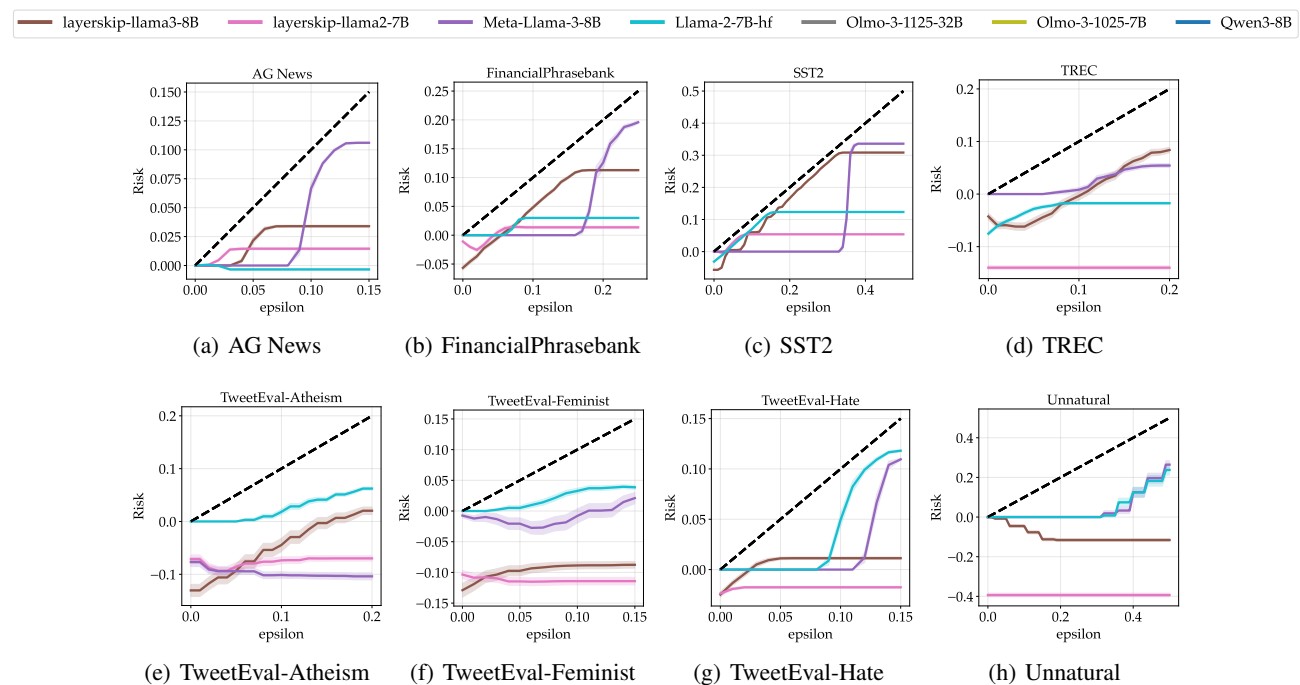

*Figure 14.* Empirical risk vs the user-specified risk level $\epsilon$ using our risk transformation approach over a set of 90% correct and 10% incorrect demonstrations.

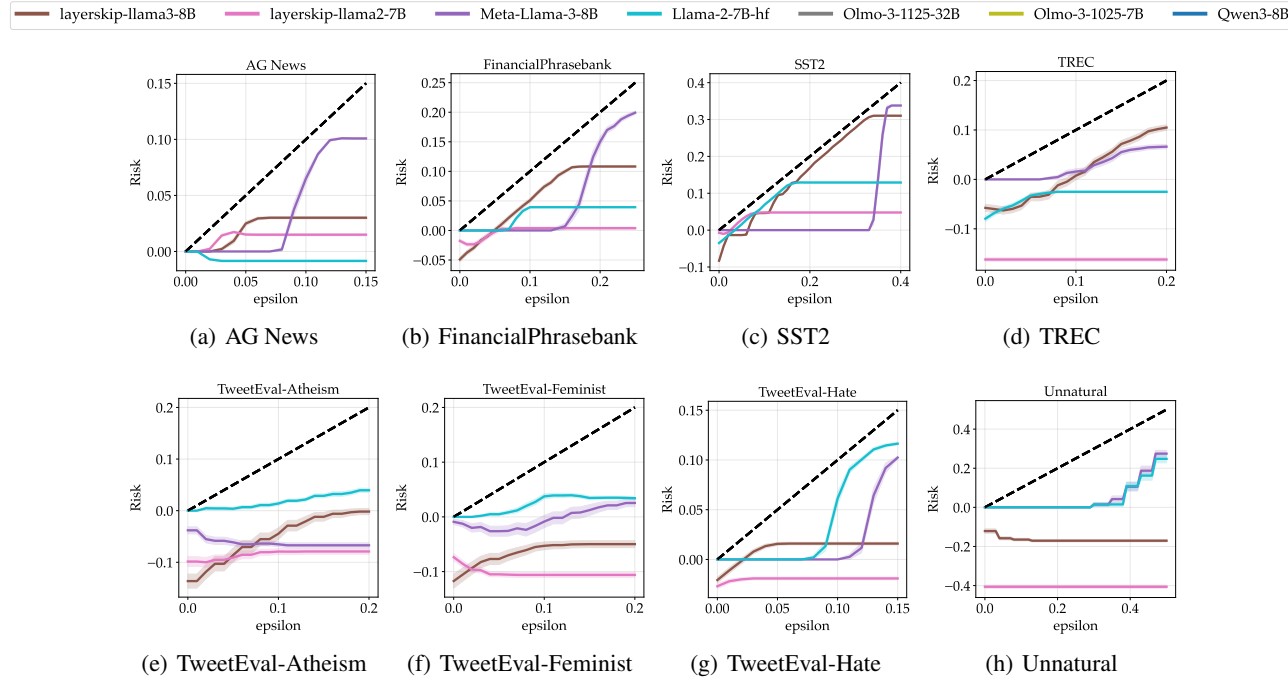

*Figure 15.* Empirical risk vs the user-specified risk level $\epsilon$ using our risk transformation approach over a set of 95% correct and 5% incorrect demonstrations.

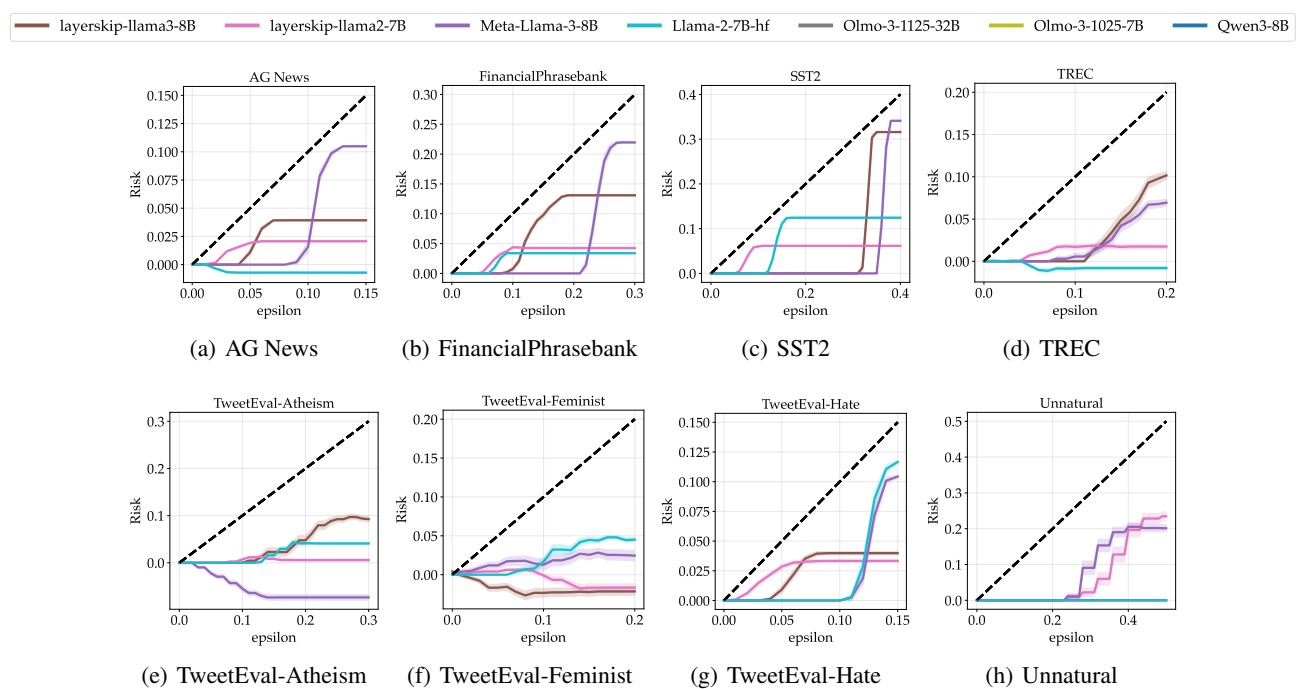

*Figure 16.* Empirical risk vs the user-specified risk level $\epsilon$ using our risk transformation approach over a set of 10% correct and 90% incorrect demonstrations.

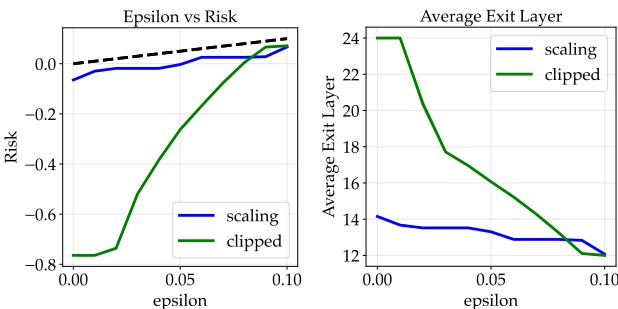

*Figure 17.* Empirical risk vs the user-specified risk level $\epsilon$ using our risk transformation approach over a set of 10% unanswerable and 90% answerable context for the SQuAD-2.0 dataset.

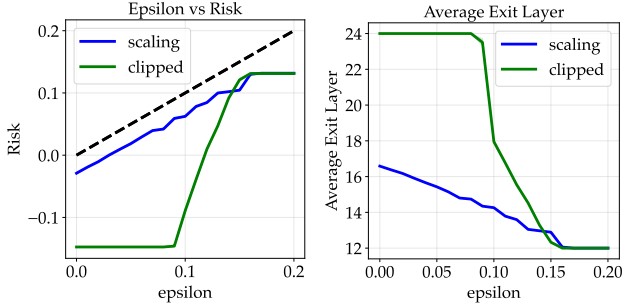

*Figure 18.* Empirical risk vs the user-specified risk level $\epsilon$ using our risk transformation approach over a set of 10% answerable and 90% unanswerable context for the SQuAD-2.0 dataset.

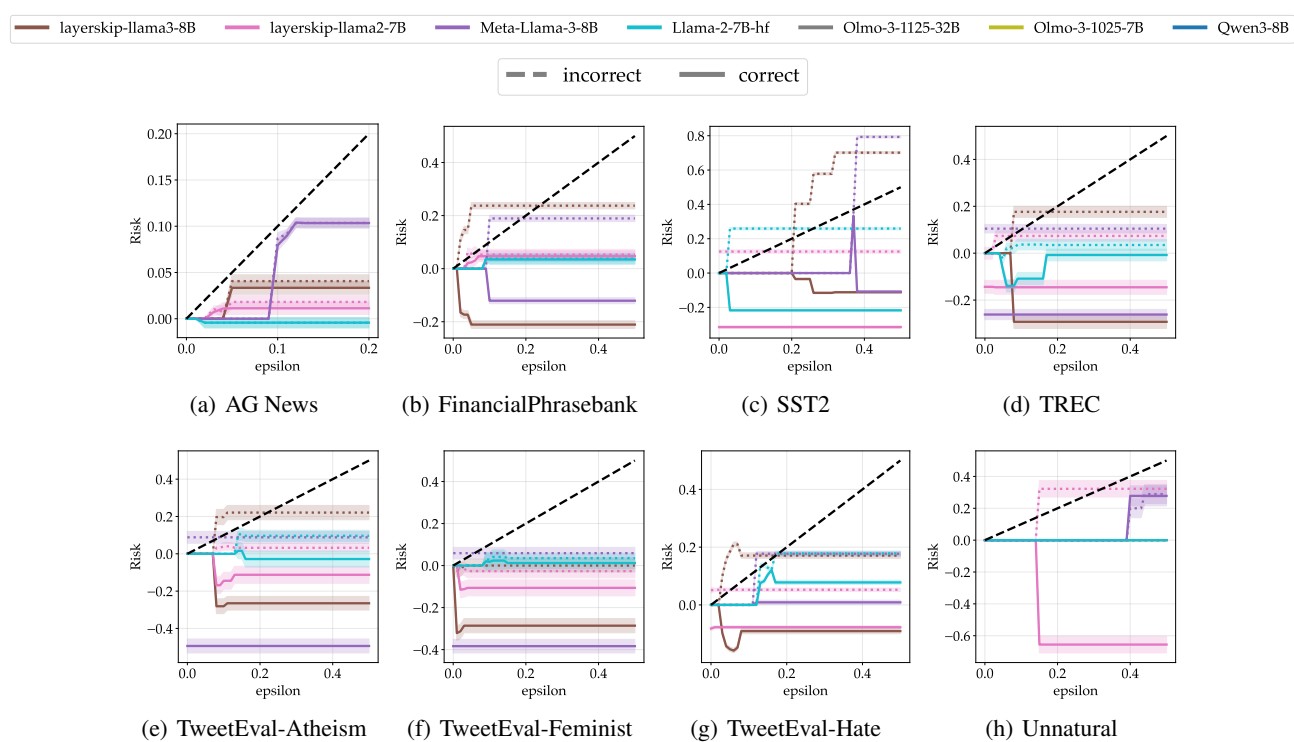

*Figure 19.* Empirical risk vs the user-specified risk level $\epsilon$ using our risk transformation approach over a set of 50% correct and 50% incorrect demonstrations. We examine the class-conditional risks for correct and incorrect demonstrations respectively. Shaded regions correspond to one standard error computed over 100 experiments.

violate the risk-control on the subclass of only incorrect demonstrations.

Ideally, we would like to condition on these sub-populations separately, but we cannot know ahead of time whether we are given correct or incorrect demonstrations. Future work should investigate how we can integrate additional control mechanisms with our approach to ensure safe behavior in *all* subgroups of safe and unsafe prompts, potentially by borrowing ideas from class-conditional conformal prediction literature (Ding et al., 2023).

## G. Ablations

We performed many ablation studies to arrive at the setup of our experiments in this paper. Details of these ablations are provided here, as well as the particular settings under which we ran our experiments.

### G.1. Contextual Calibration

Contextual calibration (Zhao et al., 2021) reduces instability arising from the specific choice of prompt format and the choice and ordering of in-context examples; it has been widely applied in recent work, including in Halawi et al. (2024). We performed ablations with and without contextual calibration, and found that contextual calibration was necessary to stabilize accuracy and confidence across the layers of the model. A plot comparing an experiment with and without contextual calibration is shown in Fig.22.

### G.2. Confidence Measures

We test different ways of measuring confidence in the model's prediction to evaluate whether this impacts our risk control approach. The three measures we use are as follows:

- *argmax*: Taking the simple argmax of the logits after applying softmax.

- *top 2*: Take the difference between the top 2 largest values of the logits after applying softmax.

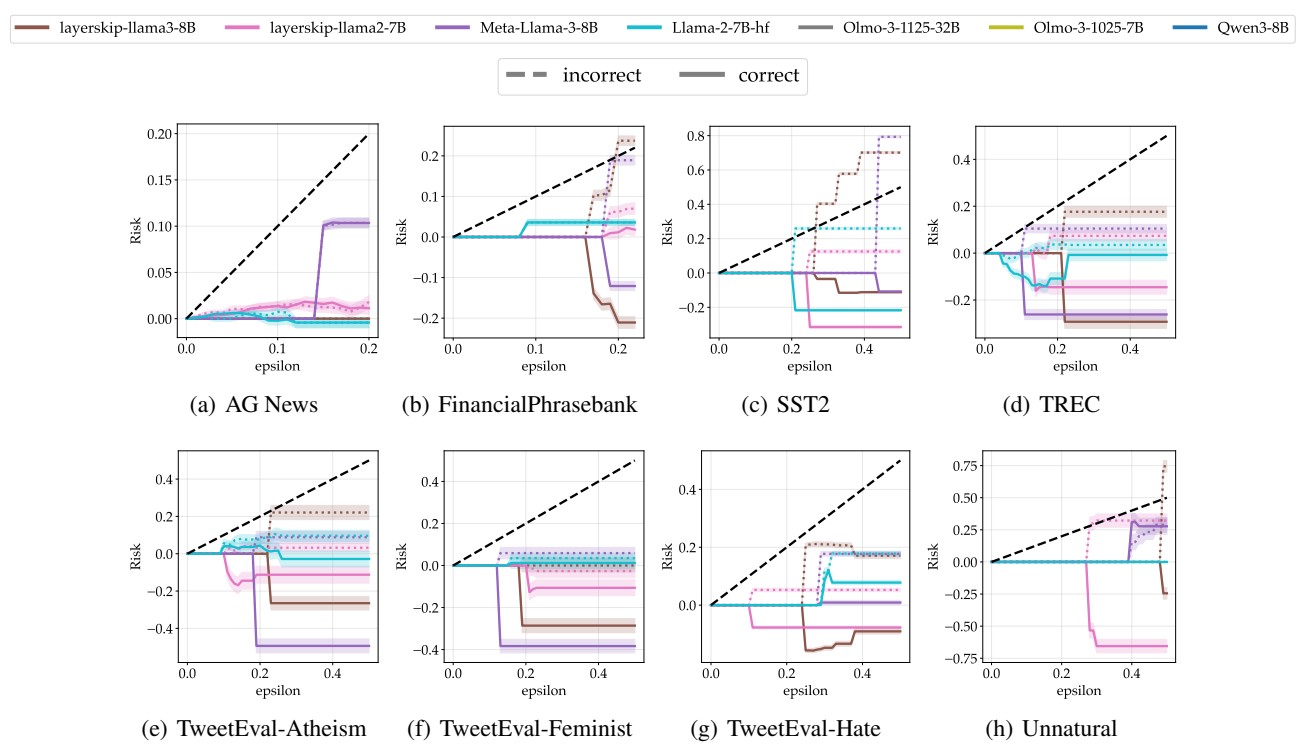

*Figure 20.* Empirical risk vs the user-specified risk level $\epsilon$ using the clipped-loss approach over a set of some correct and some incorrect demonstrations. This approach prioritizes controlling risk for incorrect demonstrations but can reduce performance gains when given correct demonstrations.

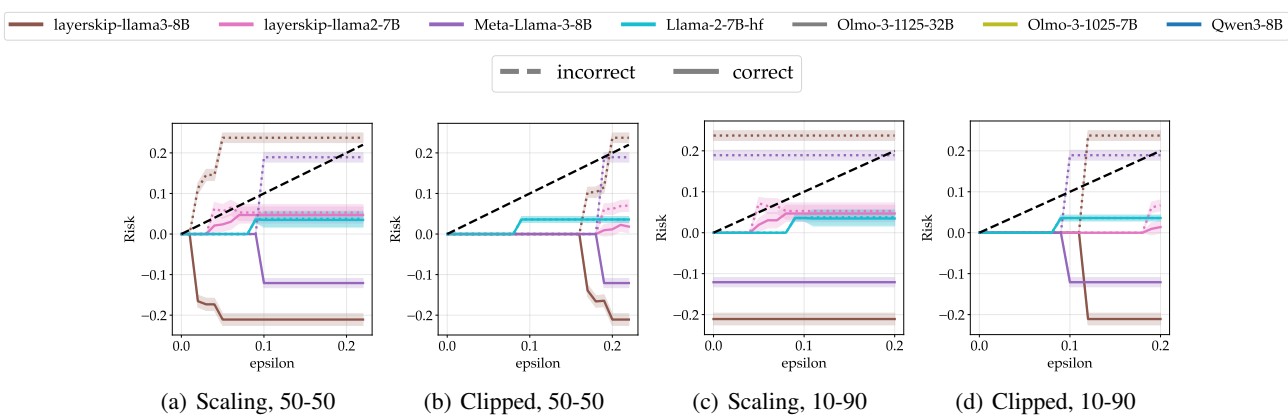

*Figure 21.* Empirical risk vs the user-specified risk level $\epsilon$ using our risk transformation approach over a set of some correct and some incorrect demonstrations. We examine the class-conditional risks for correct and incorrect demonstrations respectively on the FinancialPhrasebank dataset, where we either have a 50-50 balanced split of correct vs incorrect demos, or 10% incorrect and 90% correct demos. We find that our approach defaults to the zero-shot behavior much less often than the loss-clipping approach regardless of the proportion of correct demonstrations (as seen by the risk 0 regions for small $\epsilon$). Shaded regions correspond to one standard error computed over 100 experiments.

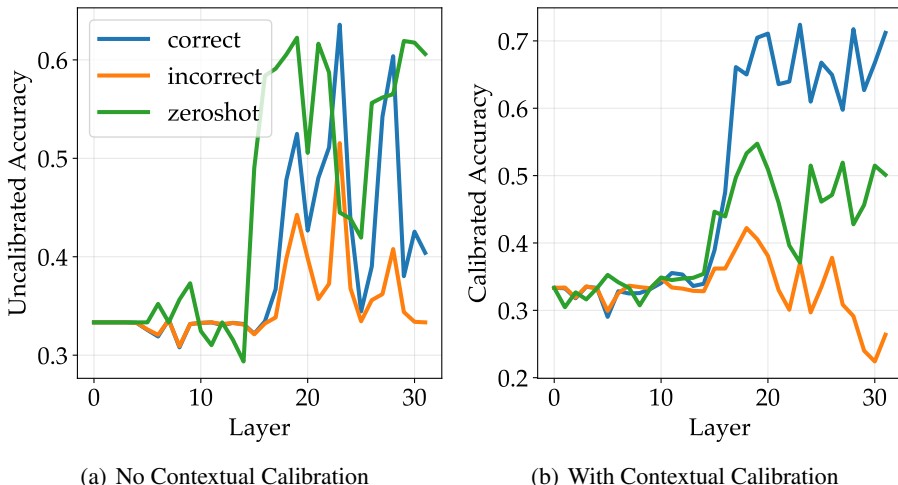

(a) No Contextual Calibration  (b) With Contextual Calibration

*Figure 22.* We find that calibration is necessary to stabilize accuracy and confidence across layers of the model. This enables effective early-exit for risk control. An example is shown here for FinancialPhrasebank.

- *entropy*: Compute the entropy over the logits post-softmax.

We present results on the TweetEval Hate dataset in Fig.23. We find that though the choice of confidence measure will affect the level of risk for specific $\lambda$ values, there is no significant impact on our risk-control approach, as it works under all scenarios. We choose *argmax* as our confidence measure for all experiments in the paper, as this is the most common approach taken in other work.

### G.3. First Exit

We find that the models are frequently overconfident in the wrong answers in earlier layers. Through detailed examination of the models' generated text from intermediate layers, we also find that the quality in very early layers is extremely low and gradually improves through the layers. This means that risk control based on model confidence will provide trivial results when we early-exit from anywhere in the model; we cannot have a confidence-based $\lambda$ threshold which allows us to early-exit while preserving performance. We address this by applying our risk-control approach only on the last half of the layers, meaning that the earliest possible exit for our 32-layer models is layer 16. Empirical results justifying this choice are shown in Fig.24.

We further note that the halfway cutoff point is an empirical choice that we show works well on our particular tasks. Different applications may require different choices of a cutoff point.

### G.4. True Dataset Labels

We show that the model has already memorized existing datasets during pre-training; results displayed in Fig.25. These results are also confirmed in prior work (Pan et al., 2023; Fang et al., 2025). This motivates our approach of transforming the task to a format that is equivalent to, but distinct from, their original form by assigning arbitrary "dummy" labels for each label of the dataset.

### G.5. Defaulting to Zero-Shot with SQuAD v2.0

For our open-ended multi-token question-answering task, SQuAD v2.0, we experimented with 3 different methods of defaulting to the zero-shot model. In contrast with the in-context learning setting, where we only had a single confidence threshold and one exit (for single-token prediction), we now have a model with multi-token generation that could exit at different layers for each token. In particular, we experimented with 1) when **all** of the tokens generated have no layer of the model which exceeds the confidence threshold $\lambda$; 2) when **any** of the tokens generated have no layer of the model which exceeds the confidence threshold $\lambda$; and 3) when the **average** confidence over all the generated tokens does not exceed the

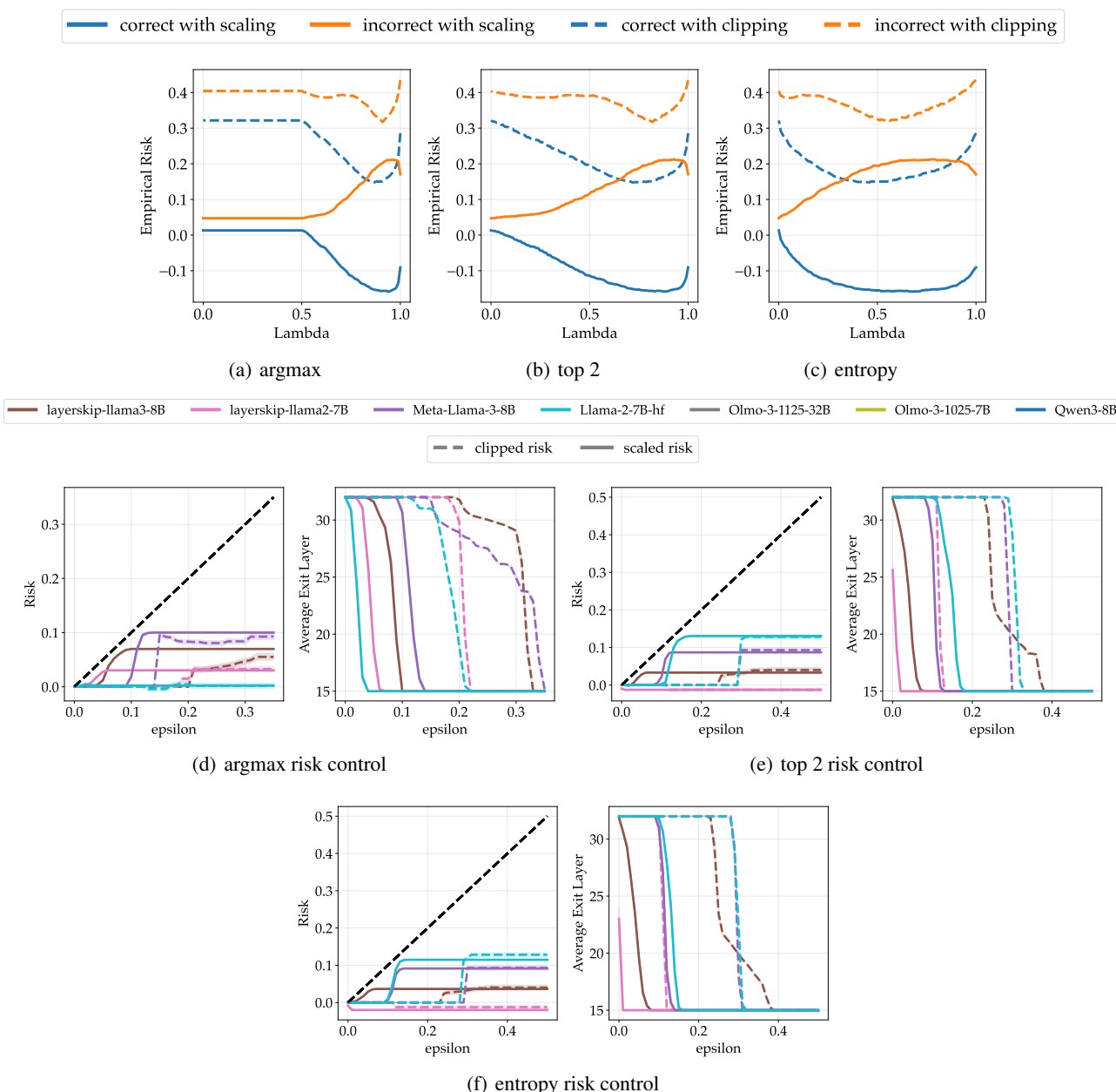

*Figure 23.* The top row shows $\lambda$ vs risk for different measures of confidence for the TweetEval-Hate dataset with the LayerSkip LLaMA 3 model, showing that different measures of confidence can impact the way that $\lambda$-thresholds on confidence affect risk (both with loss-clipping and true relative loss). The risk control plots for all models on TweetEval-Hate are shown in the last three plots; there is no significant impact on the choice of confidence measure on our risk-control approach, which works equally well on all three, without any significant differences in efficiency gains or risk level across models.

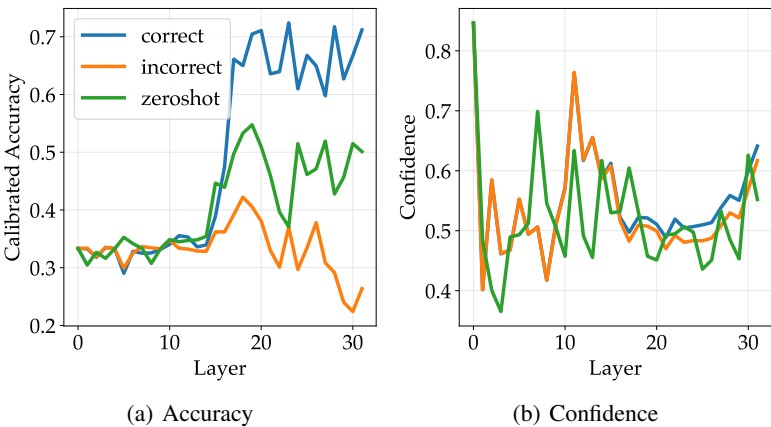

(a) Accuracy  (b) Confidence

*Figure 24.* We empirically find that the models become less overconfident and more accurate in the last half of the layers (from roughly layer 16 of 32), a finding which is consistent across all four of our models and all eight datasets. This motivates our choice to apply our early-exit risk control procedure on only the *last half* of the layers in the model. Results are shown here for FinancialPhrasebank; we found similar results across all models and datasets.

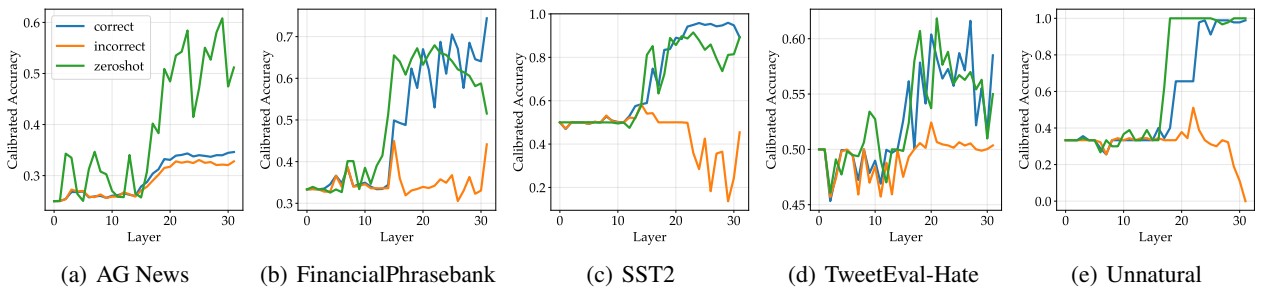

(a) AG News  (b) FinancialPhrasebank  (c) SST2  (d) TweetEval-Hate  (e) Unnatural

*Figure 25.* Here, we show the accuracy of predictions from each layer of the model. We show that our models have already memorized many of our datasets during pre-training, as shown by the fact that zero-shot will often do as well as or better than the model even given correct in-context demonstrations.

confidence threshold $\lambda$.

After performing these ablation experiments, we found that under option (1), we default to zero-shot about $1 - 2\%$ of the time on answerable questions and $2 - 3\%$ of the time on unanswerable questions. In contrast, with option (2) we default $100\%$ of the time for any nonzero threshold $\lambda$, and with option (3) we default about $80\%$ of the time for $\lambda = 0.1$, $96\%$ of the time for $\lambda = 0.2$, $99\%$ of the time for $\lambda \in [0.3, 0.5]$ and $100\%$ of the time for any threshold $\lambda > 0.5$. Because condition (1) most closely follows the proportion of examples on which zero-shot outperforms the early-exit model (about $2.5\%$ of the answerable and $20\%$ of the unanswerable questions for $\lambda = 0.5$), we found that condition (1) (**all tokens**) was the best approach.

# H. Compute Resources

To run all experiments with language models, we used 4 A100 GPUs on the Johns Hopkins DSAI compute cluster. Plots and risk-control were executed locally.

