# OpenReview forum: "Controlling the Risk of Corrupted Contexts for Language Models via Early-Exiting"
_ICML.cc/2026/Conference — ICML 2026 regular_

### Official Review · Reviewer_keVL · 2026-03-11

**Soundness:** 3
**Presentation:** 4
**Significance:** 2
**Originality:** 2
**Overall Recommendation:** 4
**Confidence:** 3

**Summary:**

This paper tackles the problem of LLMs being misled by bad user-provided context by combining early-exiting with distribution-free risk control. The key insight is that LLMs "overthink" on harmful context, deeper layers make things worse, so exiting early or falling back to zero-shot predictions can prevent corruption. The authors propose a context-aware loss measuring degradation relative to zero-shot, and a risk transformation that preserves information about helpful context while maintaining statistical guarantees. Across 9 tasks and 5 models, the approach controls risk below user-specified levels while achieving over 50% computational savings compared to prior methods.

**Compliance With Llm Reviewing Policy:**

Affirmed.

**Final Justification:**

Rebuttals answered most of my concerns.

**Key Questions For Authors:**

See the weaknesses.

**Limitations:**

See the weaknesses.

**Strengths And Weaknesses:**

__Strengths:__

1. The approach is conceptually clear and straightforward, making it intuitive to understand and practical to implement.

2. The paper is well-written and well-motivated, with a clear problem formulation.

3. The proposed framework delivers both safety guarantees and computational savings simultaneously, which makes is appealing for resource-constrained deployment scenarios.

4. The experimental setup is thorough, with 100-trial standard errors and extensive ablations on confidence measures, calibration, first-exit layers, and varying proportions of helpful versus harmful context.


__Weaknesses:__

1. The most critical weakness of this paper is that all experiments use small, underperforming models (LLaMA-2 7B, LLaMA-3 8B, T5-Large) that are far behind current state-of-the-art, raising serious doubts about generalization. In fact, larger, better-aligned models can generate more robust intermediate representations that do not degrade in deeper layers, potentially eliminating the overthinking phenomenon. Further, stronger zero-shot performance in recent models can shrink the gap that the framework aims to control to the point where it adds small value. Also, the assumption that the last half of layers is where overthinking occurs is calibrated to 32-layer models and may not transfer to architectures with different depths, widths, or designs, such as MoE. Given this, the framework risks addressing a problem that may not exist in the models that matter most today.

2. The corruption types evaluated are narrow and extreme, only fully permuted labels and adversarial unanswerable questions, while real-world corruption involves subtle errors, partially correct context, and mixed-quality information within single prompts.

3. The framework provides no class-conditional guarantees, which is a fundamental gap given that the paper's primary motivation is safety against harmful context. Risk can be well-controlled on average across the full data distribution, while the harmful-context subpopulation, which is the exact scenario the framework is designed to protect against, remains unprotected or even degraded. While this limitation has been acknowledged in the paper, I do believe that for a paper whose central contribution is safety, the inability to guarantee safety on the subgroup that matters most undermines the practical claims.

---

> ### Author Rebuttal · Authors · 2026-03-31
>
> We sincerely appreciate your thoughtful feedback. We also thank you for highlighting that our paper is “well-written and well-motivated, with a clear problem formulation” and an “intuitive [and] practical implementation”, offering “both safety guarantees and computational savings simultaneously” with “thorough [experiments] and extensive ablations.”
>
> Below we address your comments:
>
> > All experiments use small, underperforming models (LLaMA-2 7B, LLaMA-3 8B, T5-Large) that are far behind current state-of-the-art, raising doubts about generalization. Larger, better-aligned models can generate more robust intermediate representations that do not degrade in deeper layers, eliminating the overthinking phenomenon.
> >
>
> We appreciate the feedback. We repeated our experiments on all 8 ICL tasks with larger, more recent models (OLMo-3 7B and 32B and Qwen 3 8B; with 32, 64, 36 layers) and observed the following:
>
> 1. All models exhibit overthinking behavior across all 8 datasets.
> 2. In all models, the confidence became more calibrated with accuracy around the middle layer.
> 3. All models satisfy our risk control guarantees and achieve over 60% efficiency gains ($\epsilon=0.05$) with our approach, agreeing with results from section 4 and fig. 5 and 7.
>
> These results align with a broader pattern in the literature suggesting that susceptibility to harmful context induces detectable harmful internal representations even in well-aligned models in a way that persists across model types or architectures [1].
>
> > Stronger zero-shot performance in recent models can shrink the gap that the framework aims to control to the point where it adds small value.
> >
>
> We respectfully disagree: stronger zero-shot (ZS) performance will *widen* the performance gap between the safe baseline and corrupted contexts, making our framework even more effective. As long as overthinking occurs, a stronger ZS baseline simply provides a higher floor for safety while still benefitting from helpful context when available.
>
> > The assumption that the last half of layers is where overthinking occurs is calibrated to 32-layer models and may not transfer to architectures with different depths, widths, or designs.
> >
>
> Please refer to the first comment in our response to reviewer WYPy.
>
> > The corruption types evaluated are narrow and extreme, only fully permuted labels and adversarial unanswerable questions. Real-world corruption involves subtle errors, partially correct context, and mixed-quality information within single prompts.
> >
>
> Our framework was specifically engineered to move beyond these extreme binary settings by providing robust safety guarantees under *mixed-quality inputs*. Our in-context learning permuted labels setting mirrors Halawi et al [6], whose ablations (including partially-correct and random labels) provide robust justification for this approach. For SQuAD v2.0, the prompts are factually correct and high-quality; adversarial selection corresponds to selecting passages that are superficially relevant to the question, meant to mislead the model into answering a different but related question [7]. We identify an extension to mixed-quality information in the same prompt as an interesting follow-up direction to our work in the Conclusion. Our work lays the necessary principled foundation for this extension.
>
> > The framework provides no class-conditional guarantees (against harmful context).
> >
>
> Class-conditional, distribution-free risk control with finite samples is provably impossible without strong distributional assumptions [2], so this limitation is shared by all existing risk control frameworks [3,4,5]. We welcome any references to prior work that has achieved this in a similarly general setting that we can address during the discussion period. We have also expanded our discussion section to cite these impossibility results and clarify that our framework provides the strongest guarantees possible under the distribution-free regime.
>
> ### References
>
> [1] Zou et al. Improving Alignment and Robustness with Circuit Breakers. NeurIPS 2024.
>
> [2] Barber et al. The limits of distribution-free conditional predictive inference. Information and Inference: A Journal of the IMA, 2021.
>
> [3] Angelopoulos et al. Conformal Risk Control. ICLR 2024.
>
> [4] Angelopoulos et al. Learn then Test: Calibrating predictive algorithms to achieve risk control. Annals of Applied Statistics, 2025.
>
> [5] Bates et al. Distribution-free, risk-controlling prediction sets. Journal of the ACM (JACM), 2021.
>
> [6] Halawi et al. Overthinking the Truth: Understanding how Language Models Process False Demonstrations. ICLR 2024.
>
> [7] Rajpurkar et al. Know What You Don't Know: Unanswerable Questions for SQuAD. ACL 2018.

---

> > ### Author Rebuttal · Reviewer_keVL · 2026-04-04
> >
> > Thanks to the authors for their rebuttals. I have raised my score.

---

> > > ### Author Response · Authors · 2026-04-05
> > >
> > > Thank you again for the acknowledgement and for updating your score, noting that our rebuttal resolved your concerns. We greatly appreciate your time and feedback that helped us improve our paper.

---

### Official Review · Reviewer_qD2d · 2026-03-13

**Soundness:** 3
**Presentation:** 3
**Significance:** 3
**Originality:** 3
**Overall Recommendation:** 4
**Confidence:** 3

**Summary:**

This paper targets the garbage in garbage out problem where large language models get thrown off by messy or malicious context like bad in context demonstrations. The authors basically want a safety net so the model doesn't perform worse than its own zero shot baseline. They use a mix of early exiting to stop the model before it overthinks and distribution free risk control to keep the error rate under a user defined limit. A big part of the work is also a new way to scale the loss so they can handle helpful context that gives negative loss values, which usually breaks the math in standard risk control frameworks. They tested this across a bunch of classification tasks and some open ended QA and showed they can keep things safe while actually saving a lot of compute.

**Compliance With Llm Reviewing Policy:**

Affirmed.

**Key Questions For Authors:**

See weakness.

**Limitations:**

Yes

**Strengths And Weaknesses:**

## Strengths

1. Using zero-shot performance as a safe baseline is a clever, grounded way to anchor risk control.


2. The risk transformation method effectively handles negative loss, allowing the model to actually learn from helpful context.


3. Achieving over 50% computational speedup while maintaining safety guarantees is a huge practical win.


4. Extending the framework to multi-token generation on SQUAD v2.0 makes it much more relevant for real-world LLM use cases.



---

## Weaknesses

1. The calibration depends on a specific 50-50 mix of correct and incorrect demonstrations. If the deployment distribution shifts, for instance, if a user provides 100% malicious context, does the threshold $\hat{\lambda}$ calculated from a balanced mix still provide a reliable safety guarantee?


2. In the SQUAD v2.0 experiments, you generate up to 5 answers with GPT-5 to serve as ground truth for unanswerable questions. Since these aren't part of the original dataset, how much does the evaluated risk $R_c$ depend on the quality and potential biases of these LLM-generated references.


3. The method uses a single global confidence threshold $\hat{\lambda}$ for all layers. Given that LLM layers exhibit distinct processing phases, like the abstraction phase mentioned in related work, the risk control might be more effective and efficient if the exit criteria were layer-specific.


4. For multi-token generation, you default to zero-shot only if *all* generated tokens fail to meet the confidence threshold at any layer. This seems like a very loose safety trigger; if a model generates a long, harmful response but is highly confident in just one or two intermediate tokens, will it still fail to trigger the zero-shot safety fallback?

---

> ### Author Rebuttal · Authors · 2026-03-31
>
> We sincerely appreciate your thoughtful feedback. We also thank you for highlighting that our method is a “clever, grounded way to anchor risk control” which “allows the model to learn from helpful context” and is “relevant for real-world LLM use cases”, achieving “huge practical wins” via “over 50% computational speedups”.
>
> Below we address your comments:
>
> > The calibration depends on a 50-50 mix of correct/incorrect demos.
> >
>
> We appreciate the reviewer's attention to this aspect of our setup; however, we respectfully note that this does not reflect our methodology. In Section G and Figures 12–15, we demonstrate that our approach successfully controls risk across many distributions of context quality, including up to 100% helpful or harmful context (Fig. 6 & 8).
>
> > If the deployment distribution shifts, e.g. if a user provides 100% malicious context, does the threshold still provide a reliable safety guarantee?
> >
>
> The assumption that the calibration (or training) set is representative of the test-time distribution is a fundamental requirement of *all* risk control frameworks [1,2,3], and more broadly for any learning problem. As long as this i.i.d. assumption is satisfied, our method provides reliable safety guarantees regardless of the distribution of context quality; see response above.
>
> > In the SQUAD v2.0 experiments, you generate answers with GPT-5 as ground truth for unanswerable questions. How much does the risk depend on the quality and biases of these references.
> >
>
> We would like to clarify that the GPT-5 generated answers are intended to be a gold-standard zero-shot label for the unanswerable SQuAD questions, which lack standard factual ground truths. We updated the wording in our paper to reflect this. Since our experiments were run with much weaker LLMs, we can expect that GPT5’s answers will always be as good or better than the answers produced by our models; this claim is supported by other work that uses GPT-generated labels to improve weaker models [5].
>
> Further, our context-aware loss is a comparative metric; because zero-shot and early-exit predictions are evaluated against the same synthetic targets, any potential bias is applied *symmetrically* and does not invalidate the relative risk measurement. This is empirically validated by our system’s ability to discriminate context quality, as it defaults to zero-shot behavior 3.3x more frequently on misleading questions than answerable ones and achieves a performance gain in 71% of those fallback cases (see Sec. 4.3).
>
> > The method uses a single global confidence threshold for all layers. The risk control might be more effective and efficient if the exit criteria were layer-specific.
> >
>
> We appreciate the reviewer’s suggestion. This is a design choice we made because layer-specific thresholding would introduce new challenges in practice (a loss of statistical power and an exponentially larger grid search over thresholds [4]); thus, existing papers on risk control make the same choice [1,2,3]. This would be an interesting but orthogonal direction for future work, as we could still apply our approach with layer-specific thresholds.
>
> > For multi-token generation, you default to zero-shot only if *all* generated tokens fail to meet the confidence threshold at any layer. This seems like a loose safety trigger; if a model generates a long, harmful response but is highly confident in just one or two intermediate tokens, will it still fail to trigger the zero-shot safety fallback?
> >
>
> While the concern regarding a loose safety trigger is relevant for longer generations, such a scenario was unlikely for our QA task. During development, we rigorously evaluated three distinct strategies for defaulting to zero-shot: (1) when **all** tokens fail to exceed the confidence threshold at any layer, (2) when **any** token fails to exceed the threshold, and (3) when the **average** per-token confidence remains below the threshold. Our results indicated that (1) was most effective for our task, as it produced a fallback rate best aligned with the frequency at which the zero-shot baseline outperformed the adapted model. Further details on these ablations are provided in Appendix Section J.5. We agree that this trigger might require adjustment for longer generation tasks to prevent confident but isolated harmful tokens from bypassing the safeguard; we added a discussion of this approach in the paper as a design choice for our task.
>
> # References
>
> [1] Angelopoulos et al. Conformal Risk Control. ICLR 2024.
> [2] Angelopoulos et al. Learn then Test: Calibrating predictive algorithms to achieve risk control. Annals of Applied Statistics, 2025.
> [3] Bates et al. Distribution-free, risk-controlling prediction sets. Journal of the ACM (JACM), 2021.
> [4] Ringel et al. Early time classification with accumulated accuracy gap control. ICML 2024.
> [5] Yu et al. MetaMath: Bootstrap Your Own Mathematical Questions for Large Language Models. ICLR 2024.

---

> > ### Author Rebuttal · Reviewer_qD2d · 2026-04-01
> >
> > Thanks to the author for providing more information, I am maintaining the original score.

---

> > > ### Author Response · Authors · 2026-04-05
> > >
> > > Thank you again for the acknowledgement and for noting that our rebuttal resolved your concerns. We noticed that the score may not have been updated alongside that acknowledgement. If your current assessment is indeed that the concerns have been resolved, we would be very grateful if you could reconsider the score to reflect that view. In any case, we appreciate your time and feedback.

---

### Official Review · Reviewer_WYPy · 2026-03-19

**Soundness:** 3
**Presentation:** 3
**Significance:** 2
**Originality:** 2
**Overall Recommendation:** 4
**Confidence:** 2

**Summary:**

This paper studies methods to mitigate the impact of harmful context on LLM performance. Specifically, the authors integrate a zero-shot baseline into an early exit mechanism to ensure results are at least as good as the zero-shot output. Furthermore, the study employs a domain-specific risk transformation technique to search  for optimal parameters for the early exit mechanism while adapting to diverse domain characteristics.

**Compliance With Llm Reviewing Policy:**

Affirmed.

**Final Justification:**

After evaluating the paper and the authors' rebuttal, I have decided to maintain my original assessment. My decision is based on a balanced weighing of the following primary strengths and weaknesses:

**Strengths**: The paper proposes a straightforward and intuitive method to mitigate the impact of harmful context. Its integration of zero-shot baselines into an early exit mechanism is an effective way to establish a performance floor.

**Weaknesses**: A major concern remains regarding the "first exit" phenomenon. The current solution—a heuristic that disables early exits for the first half of the layers—lacks a rigorous justification. While the authors discussed this in Figure 24 and provided further clarification during the rebuttal, the existing evidence is not sufficiently compelling to establish the mechanism's overall robustness.

In summary, while the paper presents a solid practical contribution,the concerns regarding its robustness have not been fully resolved. Therefore, I maintain my final score of 4 (Accept).

**Key Questions For Authors:**

1. In Line 193, the parameter $\epsilon$ is specified to be positive. It would be beneficial if the authors could provide more intuition for this choice.

2. Please elaborate on the computational complexity of determining the optimal threshold $\hat{\lambda}$.

**Limitations:**

Please refer to the ‘Weaknesses’ part.

**Strengths And Weaknesses:**

**Strengths**

1.	The paper presents a straightforward and effective approach to shielding LLMs from the negative effects of harmful context.

2.	The method was evaluated across diverse tasks and LLMs, sufficiently demonstrating its effectiveness.

**Weaknesses**

1.	About the significance: The proposed method is vulnerable to the "first exit" phenomenon (premature overconfidence in shallow layers). The current workaround—disabling early exits for the first half of the layers—is a somewhat rigid heuristic that might benefit from a more systematic or scientific foundation to ensure reliable real-world deployment. Furthermore, as this issue is only briefly mentioned in the appendix, it would greatly strengthen the paper if the authors could bring this discussion into the main text to address it more thoroughly.

2.	About the originality: The "Domain-Preserving Risk Transformation" largely builds upon the existing Learn-then-Test (LTT) framework by applying a simple linear transformation. Since early exiting is also a well-established technique, the authors are encouraged to further clarify the theoretical or architectural novelty of their method beyond the practical integration of existing tools.

3.	Small typo: Line 140: $p_L(k|x)$ should not be in bold.

---

> ### Author Rebuttal · Authors · 2026-03-31
>
> We sincerely appreciate your thoughtful feedback. We also thank you for highlighting that we propose a "straightforward and effective approach to shielding LLMs from the negative effects of harmful context" and provide robust experimental validation.
>
> Below we address your comments:
>
> > The proposed method is vulnerable to the "first exit" phenomenon (premature overconfidence in shallow layers). The current workaround—disabling early exits for the first half of the layers—is a rigid heuristic that might benefit from a more systematic or scientific foundation.
> >
>
> In all our models, early layers are systematically overconfident in incorrect predictions, and both accuracy (with helpful/no context) and calibration improve only after the midpoint of the network. Our choice to restrict early exits to the last half of layers is not heuristic; it is grounded in empirical observations consistent across *all models and datasets* (Figure 24), including LayerSkip variants pre-trained to improve intermediate representations, T5-large with a different architecture & layer count, **and OLMo 3 7B & 32B and Qwen 3 8B (on which we reproduced Fig 24 and observed the same trend)**. This aligns with recent work that identifies a structured “abstraction phase” in intermediate layers of LLMs [1]. The midpoint threshold is therefore not arbitrary; it corresponds to the empirically observed onset of reliable intermediate representations on our tasks. We agree that this is an important aspect of our approach and have added this discussion to the main text.
>
> > The Domain-Preserving Risk Transformation largely builds upon the existing LTT framework by applying a simple linear transformation. Since early exiting is also well-established, the authors are encouraged to further clarify the novelty of their method.
> >
>
> We are the first to use risk control for potentially harmful context (which requires a new loss that reflects overthinking), and we also propose the technical extensions necessary to make risk control practical under mixed-quality context (risk transformation). Prior early-exit and risk-control methods were developed independently and for unrelated goals [2,5]; we integrate them to tackle the novel safety challenge of harmful context provided at test-time. Our risk transformation is not our primary contribution; rather, our focus on safety requires applying LTT, which needs this transformation as a sub-contribution to make risk control work in this setting. The novelty of our approach in these respects has also been acknowledged by reviewer qD2d.
>
> Further, among the early-exit literature, our work is the first to apply early-exit for mixed-quality context. The closest approach to an existing baseline for our work statically prunes the last layers [1], whereas we perform dynamic early-exiting via confidence thresholding. **Repeating experiments from Fig. 5 with a static exit layer confirms that our approach provides strictly better risk control: dynamic confidence thresholding is more granular and approximates the target risk bound much more accurately.**
>
> > Line 140: $p_L(k|x)$ should not be bold.
> >
>
> Thank you for the feedback; we removed the boldface.
>
> > $\epsilon$ is specified to be positive. Please provide intuition for this choice.
> >
>
> $\epsilon$ is the user-specified tolerance for overthinking. $\epsilon=0$ corresponds to a strict requirement that the model's expected performance must be at least as good as the zero-shot baseline;  $\epsilon<0$ means requiring a *guaranteed performance improvement* over zero-shot. By specifying $\epsilon>0$, we ensure the risk control procedure can always find a valid threshold (or default to zero-shot) to satisfy risk control guarantees even in the worst-case 100% garbage-in scenario. We made a note of this in section 2.2 to clarify this point.
>
> > Please elaborate on the computational complexity of determining the optimal threshold.
> >
>
> The computational complexity of finding the optimal threshold is concentrated in a **one-time offline calibration phase**. Threshold selection happens via hypothesis testing on each threshold $\lambda \in \Lambda$ over all $N_{cal}$ calibration data points, giving a time complexity of $O(|\Lambda| * N_{cal})$. Once this is complete, applying the pre-computed threshold often leads to significant efficiency gains (>50% speedup; line 90) as reported in our paper. At test time, this could soon offset the time taken to compute the threshold.
>
> # References
>
> [1] Cheng et al. Emergence of a High-Dimensional Abstraction Phase in Language Transformers. ICLR 2025.
>
> [2] Jazbec et al. Fast yet Safe: Early-Exiting with Risk Control. NeurIPS 2024.
>
> [3] Schuster et al. Confident Adaptive Language Modeling. NeurIPS 2022.
>
> [4] Huang et al. Multi-Scale Dense Networks for Resource Efficient Image Classification. ICLR 2018.
>
> [5] Angelopolous et al. Learn then Test: Calibrating Predictive Algorithms to Achieve Risk Control. Annals of Applied Statistics 2025.

---

> > ### Author Rebuttal · Reviewer_WYPy · 2026-04-02
> >
> > The authors' answers partially addressed my concerns. I've went through the other questions by other reviewers as well. I'll keep my score of 4 (Weak Accept).

---

> > > ### Author Response · Authors · 2026-04-05
> > >
> > > Thank you again for the acknowledgement and for noting that our rebuttal resolved your concerns. We noticed that the score may not have been updated alongside that acknowledgement. If your current assessment is indeed that the concerns have been resolved, we would be very grateful if you could reconsider the score to reflect that view. In any case, we appreciate your time and feedback.

---

### Decision · Program_Chairs · 2026-04-30

**Decision:**

Accept (regular)

**Comment:**

This paper studies how to control the risk of harmful or misleading context in language models. The proposed method uses early exiting together with distribution-free risk control to ensure that context does not degrade performance too far below a safe zero-shot baseline, while still allowing gains and efficiency improvements when context is helpful.

Reviewers agreed that the paper studies a meaningful and practical problem, appreciated the clear formulation and the broad empirical evaluation across multiple tasks and models. The rebuttal strengthened the paper by clarifying the role of the risk transformation, adding discussion of larger and more recent models, and better explaining the early-exit design choices.

Some limitations remain as follows. Reviewers noted that the method relies on a restrictive early-exit design, particularly the decision to disable exits in the first half of layers, and that the current justification is still partly empirical.

Overall, I find that the paper makes a useful contribution by formulating a practical safety objective for harmful context and showing that it can be achieved together. While the final version should clarify the scope of the guarantees and discuss the remaining limitations more explicitly, the empirical evidence and rebuttal are sufficient for me to lean toward acceptance.